# Endogenous topoisomerase II-mediated DNA breaks drive thymic cancer predisposition linked to ATM deficiency

Alejandro Álvarez-Quilón [1,3,4], José Terrón-Bautista [1,4], Irene Delgado-Sainz[1], Almudena Serrano-Benítez[1], Rocío Romero-Granados[1], Pedro Manuel Martínez-García[1], Silvia Jimeno-González [1], Cristina Bernal-Lozano[1], Cristina Quintero[1], Lourdes García-Quintanilla[1] & Felipe Cortés-Ledesma [1,2 ✉]

The ATM kinase is a master regulator of the DNA damage response to double-strand breaks (DSBs) and a well-established tumour suppressor whose loss is the cause of the neurodegenerative and cancer-prone syndrome Ataxia-Telangiectasia (A-T). A-T patients and $Atm^{-/-}$ mouse models are particularly predisposed to develop lymphoid cancers derived from deficient repair of RAG-induced DSBs during V(D)J recombination. Here, we unexpectedly find that specifically disturbing the repair of DSBs produced by DNA topoisomerase II (TOP2) by genetically removing the highly specialised repair enzyme TDP2 increases the incidence of thymic tumours in $Atm^{-/-}$ mice. Furthermore, we find that TOP2 strongly colocalizes with RAG, both genome-wide and at V(D)J recombination sites, resulting in an increased endogenous chromosomal fragility of these regions. Thus, our findings demonstrate a strong causal relationship between endogenous TOP2-induced DSBs and cancer development, confirming these lesions as major drivers of ATM-deficient lymphoid malignancies, and potentially other conditions and cancer types.

[1] Centro Andaluz de Biología Molecular y Medicina Regenerativa (CABIMER), CSIC-Universidad de Sevilla-Universidad Pablo de Olavide, Sevilla 41092, Spain. [2] Topology and DNA breaks group, Spanish National Cancer Research Centre (CNIO), Madrid 28029, Spain. [3] Present address: Lunenfeld-Tanenbaum Research Institute, Mount Sinai Hospital, Toronto, ON, Canada. [4] These authors contributed equally: Alejandro Álvarez-Quilón, José Terrón-Bautista. ✉ email: fcortes@cnio.es

Ataxia Telangiectasia (A-T) is the paradigm of human inherited disease related to the DNA damage response (DDR)[1,2]. A-T is caused by loss-of-function mutations in the *ATM* gene, and is associated with multi-systemic features, affecting brain, gonads and the immune system; the main and most debilitating symptom of the disease being progressive early-onset cerebellar ataxia. Spontaneous chromosomal instability and profound hypersensitivity to agents that induce DNA double-strand breaks (DSBs) distinguish this syndrome from other spinocerebellar ataxias[3], leading to the idea of DSBs underlying A-T symptomatology, although there is still an open debate regarding the molecular trigger of A-T, and the specific contribution of DSBs to the different aspects of the disease.

The connection of DSBs with immunodeficiency and cancer predisposition in A-T is, however, well established and properly recapitulated in the *Atm*$^{-/-}$ mouse[4]. During lymphocyte development, the variable regions of T-cell receptor (*TCR*) and Immunoglobulin (*Ig*) loci, in T and B cells respectively, are randomly rearranged in a unique combination of the multiple possible V, D (in the case of *TCRβ* and δ and *IgH*) and J coding segments present in the germline[5]. This shuffling occurs through the generation of pairs of DSBs by the RAG1-RAG2 recombinase at recombination sequence signals (RSS), and their subsequent repair by the non-homologous end-joining (NHEJ) machinery. ATM, although not completely required, facilitates V(D)J recombination, providing a molecular explanation for the relatively mild immunodeficiency characteristic of A-T patients and *Atm*$^{-/-}$ mice. The main role of ATM in V(D)J recombination is to stabilize RAG post-cleavage complexes, thus protecting DNA ends and promoting their correct use for ligation[6,7], but other functions that are not fully understood become relevant in conditions in which NHEJ is compromised[8–10].

Defects in V(D)J recombination are not only responsible for the immunological problems in A-T, but also underlie its characteristic cancer predisposition. Indeed, roughly one third of patients develop cancer, mainly lymphoma or lymphocytic leukemia, which in mice manifests as very aggressive thymic neoplasias, with characteristics of T-cell acute lymphoblastic leukemia (T-ALL)[4]. A-T and *Atm*$^{-/-}$ T-cell malignancies are frequently linked to genome rearrangements involving the *TCR* loci, strongly supporting the contribution of aberrant V(D)J recombination. Most prevalent in A-T are translocations or inversions (14;14) involving the *TCRα/δ* locus. Interestingly, this event is also frequent in mice as a t(14;12) translocation, providing thus a molecular link between ATM-deficient T-cell malignancies in human and mouse[11–14]. Furthermore, T-cell malignancies in *Atm*$^{-/-}$ mice and human share other characteristics such as chromosome 15 duplication, *Notch1* amplification and *Pten* deletion[13,14]. In the currently accepted model, deficient V(D)J recombination and checkpoint defects caused by ATM loss lead to persistent DSBs that engage in oncogenic rearrangements[15]. Aberrant V(D)J recombination, however, is unlikely to represent the single driver of oncogenic translocations, as evidenced by additional V(D)J-unrelated regions of instability, and the persistent cancer predisposition observed upon RAG deficiency[16,17]. These results strongly suggest additional sources of DSBs as relevant contributors to the ATM-deficient oncogenic translocations responsible for T-cell cancer predisposition.

In this sense, aberrant action of DNA topoisomerase II (TOP2) can constitute an important source of chromosomal breakage[18]. The physiological function of TOP2 is to solve topological problems arising from DNA metabolism[19]. To do so, it cleaves both strands of DNA to gate the passage of another DNA segment, via the formation of a catalytic intermediate, the cleavage complex (TOP2cc), in which the enzyme remains covalently linked to 5′ termini of the break. Although normally very transient, as the DNA gate is rapidly resealed after DNA passage, TOP2ccs can be stabilized and result in irreversible TOP2-induced DSBs upon conflict with cellular processes and processing of the structure by the proteasome[20]. Interestingly, this does not only occur accidentally as a consequence of errors in TOP2 catalytic cycles, but underlies the clinical efficacy of a heterogeneous group of chemotherapeutic agents collectively known as TOP2 "poisons", of which etoposide is a widely characterised paradigmatic example[21]. Despite substantial efficacy, treatment with TOP2 poisons has been traditionally linked to the development of secondary haematological malignancies characterized by specific translocations[22]. In this regard, transcription and loop extrusion, a process that organizes the genome in functional loop domains by threading chromatin through the ring-shaped cohesin complex[23–25], have been proposed as the main source of DSBs and chromosomal translocations induced by etoposide treatment[26–29]. However, the incidence and impact of endogenous TOP2-mediated lesions in the absence of treatment with TOP2 poisons and their relationship with cancer development remain to be established. Concerning this, compounds present in the diet and the environment and some forms of pre-existing DNA damage, such as nicks or abasic sites, can also poison TOP2 activity[18].

Remarkably, TOP2-induced DSBs are characterized by harbouring peptide blocks derived from the enzyme that remain covalently bound to 5′ termini through a phosphotyrosyl bond. TDP2, with its 5′-tyrosyl-DNA-phosphodiesterese activity[30,31], is the only mammalian enzyme able to directly unblock TOP2-induced DSBs for their repair[32]. Interestingly, we have previously reported that ATM is specifically involved in facilitating repair of TOP2-induced DSBs in a TDP2-independent fashion, probably by promoting alternative nucleolytic pathways for the unspecific removal of the peptide adducts[33]. TDP2 and ATM, therefore, define two independent pathways for the repair of TOP2-induced DSBs, the combined absence of which has dramatic consequences for the cellular response to these lesions.

In the present study, we generate and characterise *Atm*$^{-/-}$ *Tdp2*$^{-/-}$ double-deficient mice to address the occurrence and relevance of TOP2-mediated DSBs in vivo, and, most importantly, their possible contribution to the different aspects of A-T symptomatology. We find that TDP2 loss further aggravates the predisposition of *Atm*$^{-/-}$ mice to thymic malignancies, and show a strong colocalization between RAG and TOP2 at regions that endogenously accumulate DSBs and TOP2ccs in *Atm*$^{-/-}$ *Tdp2*$^{-/-}$ thymocytes. These results strongly suggest the co-occurrence and misrepair of RAG- and TOP2-mediated DSBs as a major driving force of thymic malignancies linked to ATM deficiency.

## Results

***Tdp2*$^{-/-}$ *Atm*$^{-/-}$ and *Atm*$^{-/-}$ mice are phenotypically similar.** Considering the strong sensitivity and repair defect of *Tdp2*$^{-/-}$ *Atm*$^{-/-}$ double-deficient cells to TOP2-induced DSBs[33], we decided to address the physiological impact of these lesions in the context of the entire organism. We reasoned that, if TOP2 damage is mediating deleterious effects in the *Atm*$^{-/-}$ mouse, these would be further aggravated by *Tdp2* loss. Mice from *Tdp2* and *Atm* double heterozygote crosses were born at the expected Mendelian proportions and not displaying gross abnormalities (Supplementary Fig. 1a). Since *Atm* loss causes marked growth retardation[4], we measured body weight of male mice at 4 weeks of age (Supplementary Fig.1b). As expected, *Atm*$^{-/-}$ mice showed a more than 20% reduction in size compared to wild-type animals. Deletion of *Tdp2*, however, did not have an impact on its own, as previously reported[32], neither did it aggravate the growth retardation of *Atm*$^{-/-}$ mice. Furthermore, neither *Tdp2*$^{-/-}$ *Atm*$^{-/-}$

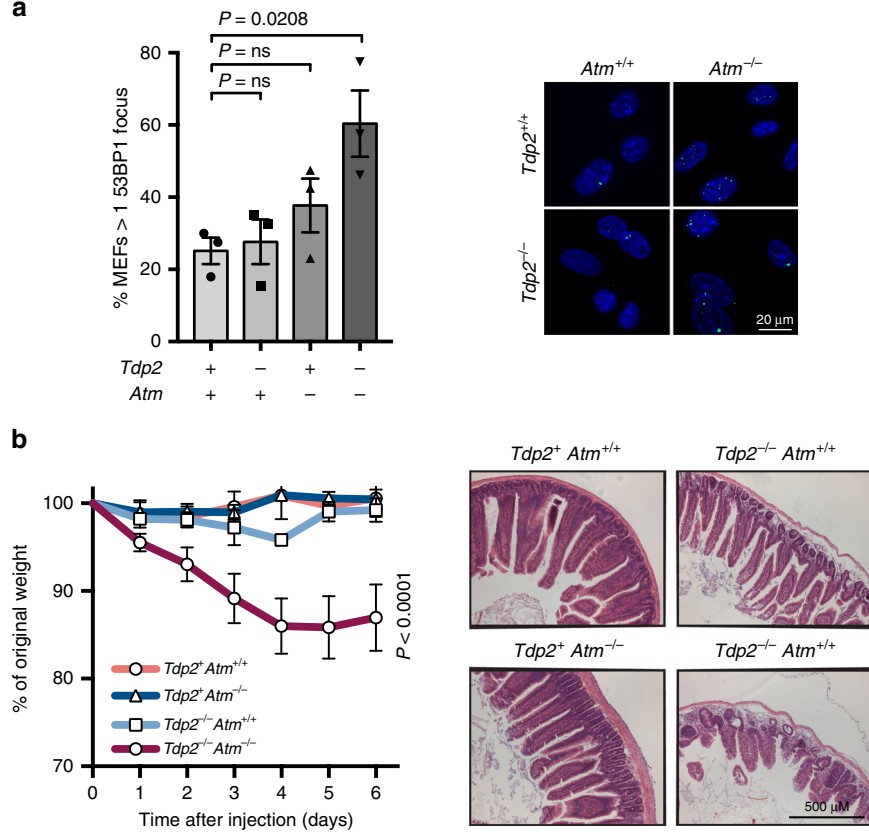

**Fig. 1 $Tdp2^{-/-}$ $Atm^{-/-}$ causes the spontaneous accumulation of DSBs and etoposide hypersensitivity in mice.** **a** Endogenous DSB occurrence in $Tdp2^{+/+}$ $Atm^{+/+}$, $Tdp2^{-/-}$ $Atm^{+/+}$, $Tdp2^{+/+}$ $Atm^{-/-}$ and $Tdp2^{-/-}$ $Atm^{-/-}$ primary MEFs measured by 53BP1 foci formation. The percentage of cells with more than one focus (left) and representative images (right) of 53BP1 foci (green) and DAPI staining (blue) are shown for each genotype. MEFs isolated from three different embryos were analysed per condition. Mean ± SEM and statistical significance by one-way ANOVA with Bonferroni post-test is shown ($F = 5,402$). **b** Body weight of 8-week old mice cohorts intraperitoneally injected with a single dose of etoposide (25 mg/kg). Average ± SEM of the percentage of initial body weight from five mice and statistical significance by Two-way ANOVA with Bonferroni post-test is shown ($F = 29,11$). $Tdp2^{+}$ indicates both $Tdp2^{+/+}$ and $Tdp2^{+/-}$ genotypes. Representative images of Haematoxylin-Eosin stained jejunum slices obtained 6 days after etoposide exposure are shown.

nor, as previously reported[4], $Atm^{-/-}$ mice displayed clear symptoms of neurological deficiencies or an ataxic behaviour, which we confirmed by analysing cerebellar integrity and Purkinje cellular density in 8-week old mice (Supplementary Fig. 1c). Finally, $Tdp2^{-/-}$ mutation did not substantially modify the already strong disruption of spermatogenesis found in $Atm^{-/-}$ mice (Supplementary Fig. 1d).

Primary $Tdp2^{-/-}Atm^{-/-}$ mouse embryonic fibroblasts (MEFs), however, showed a marked reduction in proliferation compared to single-mutant and wild-type cells (Supplementary Fig. 1e), and spontaneously accumulated DSBs in unchallenged growth conditions (Fig. 1a), with 60% of $Tdp2^{-/-}Atm^{-/-}$ cells harbouring more than one 53BP1 focus compared to 25% in wild-type. Overall, these results suggest an accumulation of endogenous DSBs in the absence of TDP2 and ATM which, at least under certain growth conditions, can disrupt cellular fitness. The in vivo incidence of these lesions, however, does not seem sufficiently high to seriously compromise tissue homeostasis during development and early stages of life.

**$Tdp2^{-/-}$ $Atm^{-/-}$ mice are hypersensitive to TOP2-induced DSBs.** Given the results above, we decided to challenge $Tdp2^{-/-}$ $Atm^{-/-}$ mice with an increased load of TOP2-induced lesions. To do so, 8-week old adult mice of the relevant genotypes were subjected to a single intraperitoneal injection of the TOP2-poison etoposide (25 mg/kg). As can be seen in Fig. 1b, $Tdp2^{-/-}$ $Atm^{-/-}$

double-knockout mice showed severe hypersensitivity to etoposide exposure, suffering a progressive weight loss that reached, on average, a 15% decrease after 6 days. Histopathological analysis revealed marked villous atrophy in the small intestine mucosa, which is a known etoposide target, as the likely cause of the drastic weight loss in $Tdp2^{-/-}$ $Atm^{-/-}$ (Fig. 1b). In contrast, at this low etoposide concentration, wild-type and $Atm^{-/-}$ animals did not respond negatively to the treatment, while $Tdp2^{-/-}$ did so only mildly. This synergistic hypersensitivity of $Tdp2^{-/-}$ $Atm^{-/-}$ mice to etoposide suggests a protective role of ATM against the adverse effects of TOP2 lesions in vivo, but only when TDP2 is absent, which recapitulates cellular observations in the context of the entire organism[33].

**TDP2 loss increases thymic cancer predisposition of $Atm^{-/-}$.** We decided to analyse $Tdp2^{-/-}$ $Atm^{-/-}$ mice over a longer period, to allow an accumulation of a higher load of TOP2-mediated damage and/or the development of potential pathologies. For this, weight and general health status were monitored weekly in a minimum of 20 mice for each of the relevant genotypes, until reaching the experimental endpoint of 2 years (730 days) (Fig. 2a). As previously reported, we observed that $Atm^{-/-}$ mice showed a reduced life-span with an important number of animals succumbing to thymic neoplasia after the first few months of life (Fig. 2b, Supplementary Fig. 2a). Furthermore, a molecular characterization of the cellular content in these tumours showed a

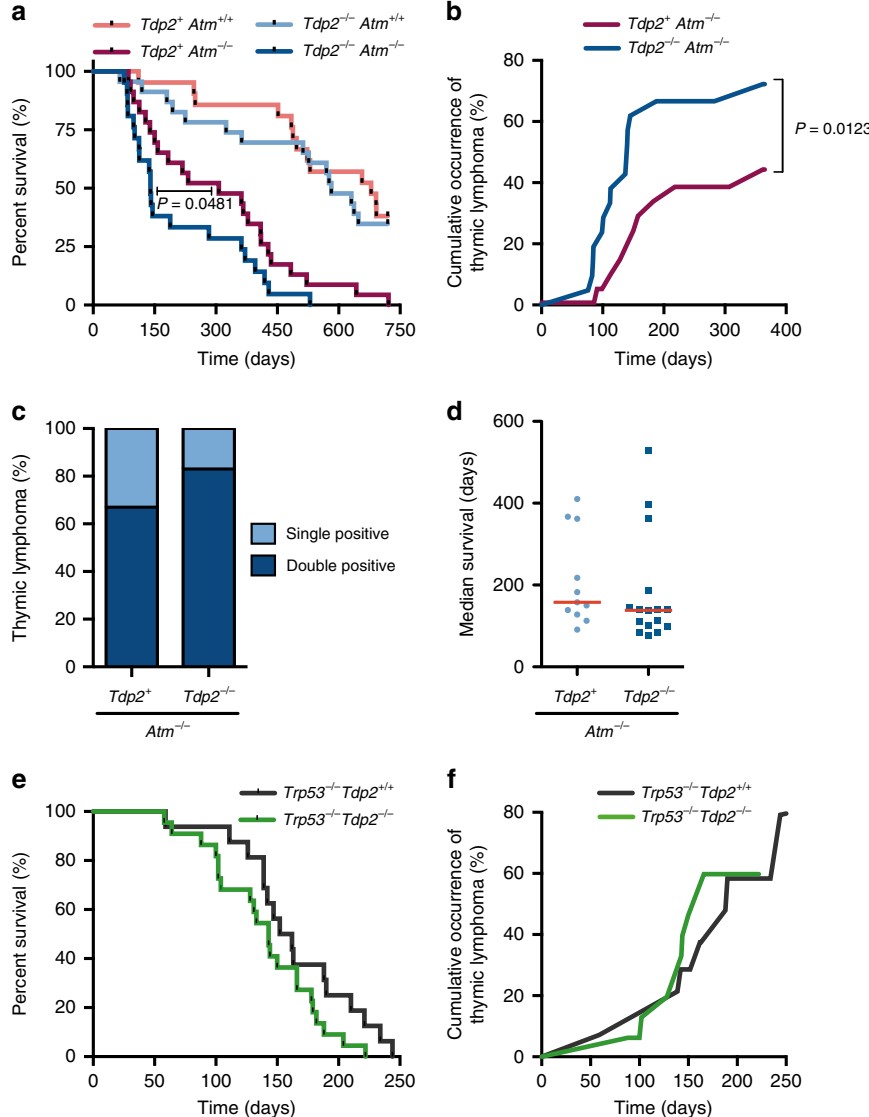

**Fig. 2 TDP2 deficiency increases thymic cancer predisposition of *Atm*$^{-/-}$ but not of *Tpr53*$^{-/-}$ mice. a, b** Kaplan–Meier survival curve (**a**) and cumulative occurrence (**b**) of thymic tumours during the 1$^{st}$ year of life of at least 20 mice with the indicated *Tdp2* and *Atm* genotypes. $n = 21$ (*Tdp2*$^{+/+}$ *Atm*$^{+/+}$), $n = 23$ (*Tdp2*$^{-/-}$ *Atm*$^{+/+}$), $n = 23$ (*Tdp2*$^{+/+}$ *Atm*$^{-/-}$), $n = 21$ (*Tdp2*$^{-/-}$ *Atm*$^{-/-}$). *Tdp2*$^{+}$ indicates both *Tdp2*$^{+/+}$ and *Tdp2*$^{+/-}$ genotypes. Statistical significance by two-sided Wilcoxon test is indicated. **c** Percentage of tumours composed by immature CD4+ CD8+ double positive, or CD4+/CD8+ single positive lymphocytes in the indicated genotypes. Thymocytes were selected by size and complexity prior classification by high or low CD4 and CD8 markers (see Supplementary Fig. 1a for gating strategy). **d** Median life-span of *Tdp2*$^{+/+}$ *Atm*$^{-/-}$ and *Tdp2*$^{-/-}$ *Atm*$^{-/-}$ mice affected by a thymic tumour ($n = 11$ and $n = 16$ for *Tdp2*$^{+/+}$ *Atm*$^{-/-}$ and *Tdp2*$^{-/-}$ *Atm*$^{-/-}$ mice respectively). **e** Kaplan–Meier survival curve and **f** cumulative incidence of thymic tumours of the indicated *Tdp2* and *Trp53* genotypes ($n = 16$ for *Tdp2*$^{+/+}$ and $n = 22$ for *Tdp2*$^{-/-}$).

majority (70%) of double positive CD4+ CD8+ T-cell tumours (Fig. 2c, Supplementary Fig. 2b), consistent with the reported T-ALL predisposition.

Single *Tdp2*$^{-/-}$ mice had a lifespan comparable to wild-type (Fig. 2a), and none of the animals developed thymic lymphoma within the course of the experiment (2 years). Strikingly however, in *Atm*$^{-/-}$ background, both the reduced overall life-span and the incidence of thymic tumours were largely aggravated by *Tdp2* inactivation (Fig. 2a, b). Thus, *Tdp2*$^{-/-}$*Atm*$^{-/-}$ mice showed a median survival of 140 days, contrasting with 307 days for the *Atm*$^{-/-}$ single mutant (Fig. 2a), and the probability of developing a thymic tumour during the 1$^{st}$ year of life increased from a 43% in *Atm*$^{-/-}$ to a 72% in the *Tdp2*$^{-/-}$*Atm*$^{-/-}$ double mutant (Fig. 2b). Importantly, lifespan of *Tdp2*$^{-/-}$*Atm*$^{-/-}$ and *Atm*$^{-/-}$ mice that did develop a thymic tumour was not significantly different (Fig. 2d), suggesting a direct effect on the incidence and not on the latency or

aggressiveness of T-cell malignancies. In this sense, tumours observed in *Tdp2*$^{-/-}$*Atm*$^{-/-}$ mice were also predominantly formed by double positive CD4+ CD8+ T-cell precursors (80%) (Fig. 2c). Double-knockout mice, however, did not show differential distribution in the populations of double negative (CD4−CD8−), double positive (CD4+CD8+) or single positives (CD4+ or CD8+) thymocytes compared to *Atm*$^{-/-}$ mice (Supplementary Fig. 2c), suggesting that the increased incidence of T-cell malignancy caused by TDP2 loss is unlikely to reflect further disruption of V(D)J recombination. In the same way, pre-B-cell, pro-B-cell and immature B-cell populations did not differ between single *Atm*$^{-/-}$ and double *Tdp2*$^{-/-}$*Atm*$^{-/-}$ knock-out mice (Supplementary Fig. 2d), further supporting the idea that V(D)J recombination is not substantially altered by TDP2 loss.

The checkpoint and proapoptotic functions of ATM, which are strongly signalled through the p53-dependent pathway, are

known to be determinant for $Atm^{-/-}$ cancer predisposition[15]. Consistent with this, p53-deficient $Trp53^{-/-}$ mice are also highly susceptible to the development of double positive CD4+CD8+ thymic tumours[34]. In order to discern between the contribution of repair and checkpoint/apoptotic functions of ATM in the increased incidence of thymic tumours in $Tdp2^{-/-}Atm^{-/-}$, we analysed life span and tumour incidence in $Tdp2^{-/-}Trp53^{-/-}$ animals, in which the repair component is not substantially affected. In contrast to what was observed in $Atm^{-/-}$ background (Fig. 2a, b), loss of functional TDP2 did not significantly decrease lifespan (Fig. 2e), nor increase the high incidence of cancer (Supplementary Fig. 2e), and thymic lymphoma in particular (Fig. 2f), observed in $Trp53^{-/-}$ animals. This suggests that $Tdp2^{-/-}Atm^{-/-}$ tumours likely reflect direct functions of ATM in the repair of TOP2-induced DSBs, and not only in their signalling for checkpoint and apoptosis. Overall, the results are consistent with a model in which misrepair of TOP2-induced DSBs can strongly contribute to the development of thymic tumours characteristic of ATM deficiency.

**$Tdp2^{-/-}Atm^{-/-}$ and $Atm^{-/-}$ malignancies are molecularly similar**. In order to gain insights into the molecular events responsible for thymic tumour development in $Tdp2^{-/-}Atm^{-/-}$ mice, we analysed and compared copy-number variation by comparative genomic hybridization (CGH) in six $Tdp2^{-/-}Atm^{-/-}$ and three $Atm^{-/-}$ thymic tumours (Fig. 3a, Supplementary Fig. 3 and Supplementary Table 1). In $Atm^{-/-}$ animals (Fig. 3a, b), we observed the previously reported features of genomic instability[13], with amplification upstream of the $Tcra/d$ locus in chromosome 14 (2 out of 3 mice) and variable hemizygous loss of a $Bcl11b$-containing telomeric region of chromosome 12 (all 3 mice), which are indicative of the frequent t(14;12) translocation and subsequent breakage-fusion-bridge cycles. In addition, one mouse displayed instability at the $Tcrb$ locus, with both gain and loss of DNA sequence. Two of the three animals also presented the characteristic duplication of chromosome 15. Also, two animals presented deletions in chromosome 19 covering the $Pten$ tumour suppressor, although only in one of the cases this corresponded to the previously reported homozygous loss[13]. Finally, we did not observe amplification of $Notch1$ in chromosome 2, which is another common, although less frequent, feature in $Atm^{-/-}$ T-cell malignancies.

Surprisingly, we observed very similar features in $Tdp2^{-/-}Atm^{-/-}$ mice (Fig. 3a, b). Four out of six tumours presented $Tcra/d$ linked amplification, and all six showed chromosome 12 deletion with $Bcl11b$ loss. Instability at the $Tcrb$ locus was also observed in two of the six cases. Trisomy of chromosome 15 was also observed in four of the six animals, while only one of the six cases carried $Notch1$ amplification. All these events were coincident with the results obtained in $Atm^{-/-}$ animals (see above) and previous reports[13]. Finally, as observed for $Atm^{-/-}$, one $Tdp2^{-/-}Atm^{-/-}$ animal displayed hemizygous $Pten$ loss as part of a larger deletion. The variable length of the copy-number variation events at common regions, together with additional sites of instability that are unique to individual tumours (Supplementary Fig. 3), are indicative of the stochastic nature of the oncogenic events. In summary, we can conclude that loss of TDP2 increases the incidence of the same oncogenic rearrangements that drive thymic tumour in $Atm^{-/-}$ animals, suggesting a strong contribution of TOP2-mediated lesions to ATM-deficient T-cell cancer predisposition.

**TOP2B is enriched at sites of DSB accumulation**. In order to further establish aberrant TOP2 activity as a direct contributor to $Atm^{-/-}$ oncogenic T-cell genome rearrangements, we decided to map binding of topoisomerase IIß (TOP2B), which is the main source of TOP2 activity in G1 and non-cycling cells. ChIPseq analysis in freshly isolated thymocytes from wild-type mice revealed a pattern of TOP2B distribution consistent with that previously found in other cell types and tissues[35–37] (Fig. 4a), with a clear enrichment at promoter, enhancer and insulator regions (34%, 19% and 13% of TOP2B peaks, respectively). In fact, regardless of the type of functional element analysed, and as previously reported[36], we observed a strong correlation between TOP2B and the core cohesin subunit RAD21, with 81% of TOP2B peaks overlapping with those of cohesin (Fig. 4b), and detectable cohesin signal in virtually all TOP2B peaks, regardless of the presence of RNA polymerase II (POLR2A) or the insulating architectural protein CTCF (Fig. 4c). Fig. 4d shows the $Cxcr4$ locus merely as an illustrative example of TOP2B and cohesin co-localization within promoter (left) and insulator (right) regions. These results are consistent with the proposed connection between TOP2B function and 3D-genome organisation[36,37].

We decided to specifically study TOP2B occupancy at particular sites related to $Atm^{-/-}$ oncogenic translocations, integrating published maps of endogenous DSB accumulation determined by ENDseq in $Atm^{-/-}$ thymocytes [38]. When concentrated on the $Bcl11b$, $Notch1$, and $Pten$ loci (Fig. 5a–c), we observed clear TOP2B binding at all of these regions with strong accumulation at the promoters and towards the 3' end of the genes, with a pattern following that of cohesin and RNA-polymerase II (POLR2A). This TOP2B accumulation was not thymocyte-specific, as a similar pattern was observed in MEFs (shown for $Bcl11b$, Supplementary Fig. 4a). In order to directly identify regions of DSB accumulation in $Atm^{-/-}$ thymocytes, we selected ENDseq peaks specific for $Atm^{-/-}$ thymocytes (highlighted in yellow; Fig. 5a–c). Interestingly, when these sites were compared with the distribution of TOP2B, overlap was highly variable with examples of both unique and common regions; compare region 1 and region 2 in $Pten$ (Fig. 5c). TOP2B is therefore present at $Atm^{-/-}$ unstable regions, but it is difficult to establish a direct association with DSB accumulation, and may be a mere consequence of its widespread distribution at regulatory regions. Furthermore, the difficulty to assign original translocation breakpoints with sufficient precision strongly limits our capacity to directly link TOP2B binding to oncogenic DSB occurrence at these specific locations.

Based on this, we decided to compare the distribution of TOP2B and ENDseq signal at a genome-wide level. We observed a clear accumulation of DSBs at TOP2B peaks (Fig. 5d), supportive of a link between TOP2B function and spontaneous chromosome fragility, in the same line as what has been previously reported for etoposide treatments[37]. Conversely, and further supportive of this TOP2B-DSB association, ENDseq peaks displayed an accumulation of TOP2B signal (Fig. 5e). Most prominent peaks of ENDseq signal in thymocytes are known to be associated with RSS sites, disappearing in $Rag2^{-/-}$ and strongly accumulating in $Atm^{-/-}$ cells[38], consistent with the high incidence and precision of RAG cleavage and the reported roles of ATM in facilitating V(D)J recombination. At TOP2B sites, however, DSBs were noticeably increased in both $Rag2^{-/-}$ and $Atm^{-/-}$ thymocytes (Fig. 5d), demonstrating that they are unrelated to RAG cleavage, and suggesting that these lesions can accumulate upon ATM deficiency. In summary, there is a significant genome-wide correlation between sites of TOP2B function and occurrence of RAG-independent endogenous DSBs in thymocytes, suggesting aberrant TOP2B activity as an additional relevant source of chromosome breakage that could contribute to characteristic $Atm^{-/-}$ oncogenic translocations.

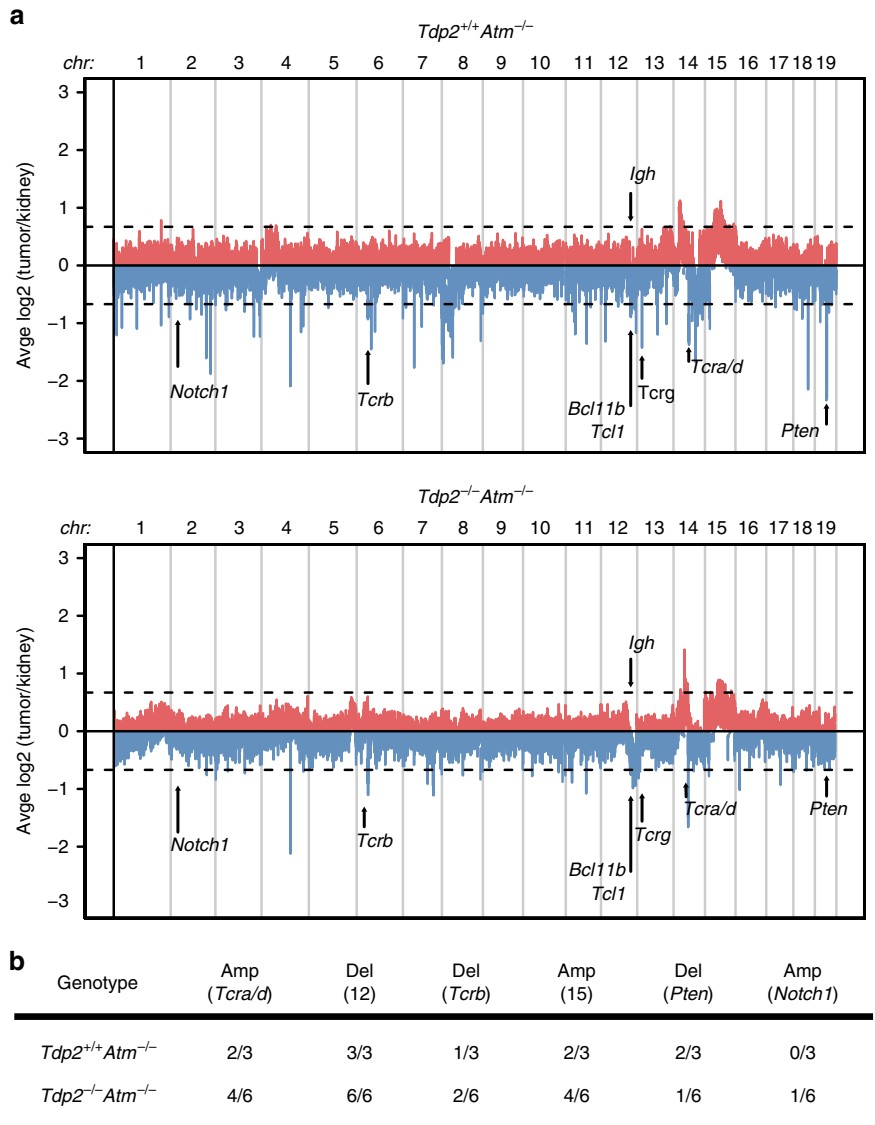

**Fig. 3 *Tdp2⁻/⁻Atm⁻/⁻* and *Atm⁻/⁻* thymic malignancies display similar genome rearrangements. a** Merged CGH analysis of *Tdp2⁺/⁺Atm⁻/⁻* (three mice, top) and *Tdp2⁻/⁻Atm⁻/⁻* (six mice, bottom) thymic lymphomas. DNA from each tumour sample was hybridized and analysed using kidney DNA from the same mouse as a control. Average amplification (red) or deletion (blue) score (Log2 tumour/kidney ratio) is shown. Significant copy number variations are defined by −0.66>Log2 tumour/kidney>0.66 (dashed lines). The location of relevant loci in *Atm⁻/⁻* thymic tumours is indicated. **b** Table summarising the number of mice of each genotype displaying copy number variation at the indicated locus of interest. Amp(*Tcra/d*): amplification upstream of *Tcra/d*; Del(12): deletion of the telomeric region of chromosome 12 covering *Bcl11b*; Del(*Tcrb*): deletion at the *Tcrb* locus; Amp(15): trisomy of chromosome 15; Del(*Pten*): deletion at the *Pten* locus; Amp(*Notch1*): duplication at the *Notch1* locus.

**TOP2B and cohesin colocalize with the RAG endonuclease.** Based on the results above, we decided to check how RAG and TOP2B contribute to DSB accumulation in more detail. In order to do so, we analysed TOP2B, RAD21 and ENDseq signal in previously reported peaks of RAG1 and RAG2[12] (Fig. 6a–c). RAG2 has a wide distribution with a strong presence at active promoters displaying the H3K4me3 histone mark, while the presence of RAG1 is more restricted, only co-localizing with a fraction of RAG2 peaks[39,40]. We observed a striking genome-wide co-localization of TOP2B and RAD21 with RAG2, which was irrespective of the strong presence of RAG1 or not (Fig. 6a). Thus, clear TOP2B accumulation was observed in peaks containing both RAG1 and RAG2 but also in those containing mostly RAG2 (Fig. 6a, b). In addition, 42% of TOP2B peaks colocalized with RAG (merge of RAG1 and RAG2 peaks) (Fig. 6b), consistent with their mutual enrichment at H3K4me3-positive promoter regions.

In any case, although some ENDseq signal was observed at these regions, the accumulation of DSBs was not clear or sufficiently localized (Fig. 6a). Interestingly, when directly compared, spontaneous DSB accumulation was more robust at TOP2B and cohesin than at RAG1 or RAG2 peaks, both in wild-type and *Atm⁻/⁻* thymocytes (Fig. 6d). Altogether, these results uncover a surprising genome-wide connection between RAG and TOP2B-cohesin, and suggest that, when analysed globally, TOP2B activity can constitute an important source of DNA breakage in *Atm⁻/⁻* thymocytes, quantitatively more relevant than DSBs directly induced by RAG.

Based on this genome-wide colocalization between TOP2B and RAG, we decided to check the distribution of TOP2B specifically at sites actively undergoing V(D)J recombination in thymocytes (Fig. 7a–c; Supplementary Fig. 5a, b); this includes the *Tcra/d*, *Tcrb*, *Tcrg* and *IgH*[15] loci. Strikingly, sites of strong and localized

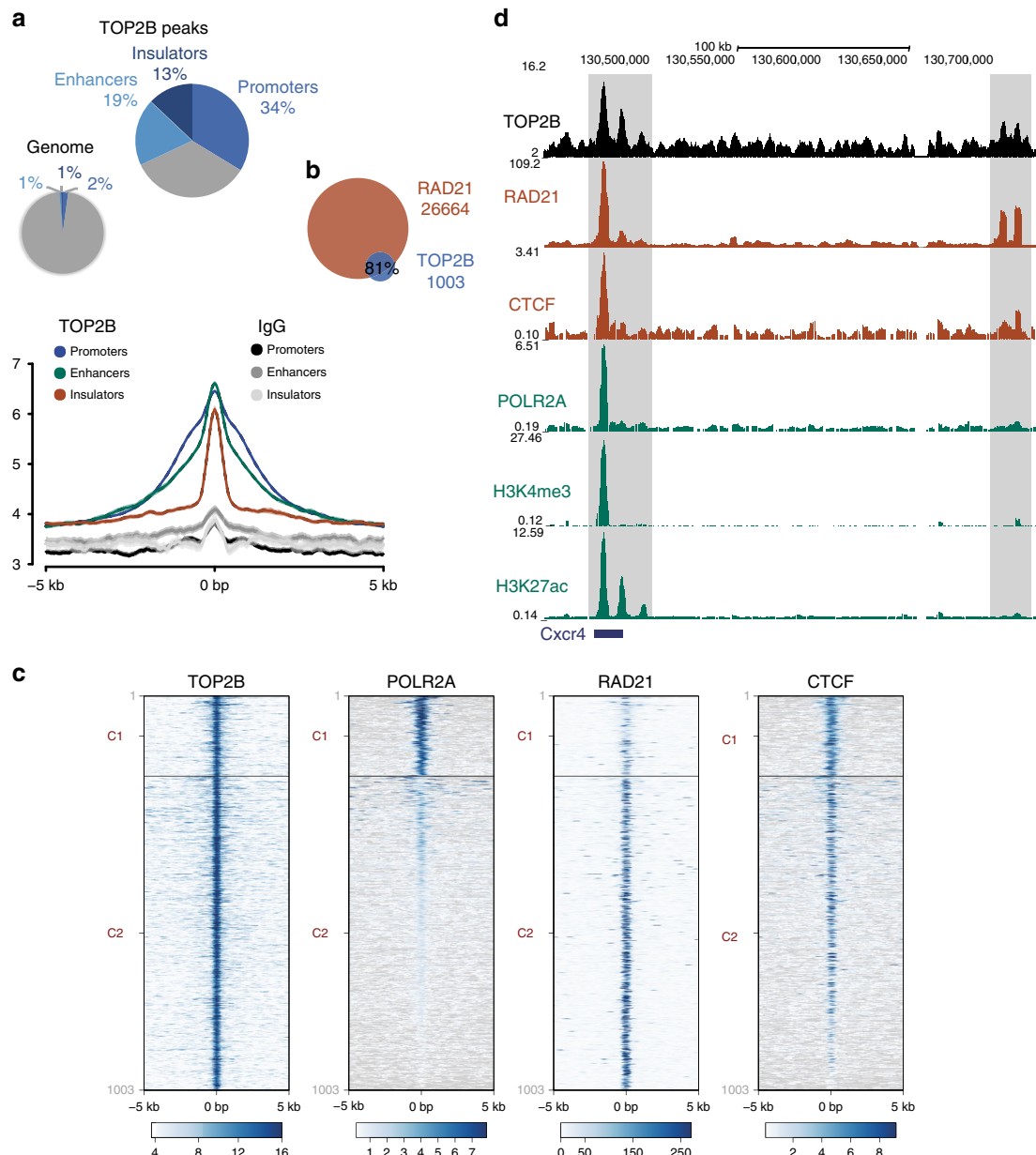

**Fig. 4 TOP2B colocalizes with cohesin at promoter, enhancer and insulator regions in thymocytes. a** Genome-wide distribution of 1003 TOP2B ChIPseq called peaks at different genomic features (promoters, enhancers, insulators and others) in wild-type primary thymocytes. Chromatin states were defined using ENCODE ccREs registry: H3K4me3, TSS and low H3K27ac for promoters, H3K27ac and low H3K4me3 for enhancers and high CTCF and no presence of the previous histone marks for insulator-like regions.TOP2B peak distribution compared to the entire genome (top), and global signal profile compared to IgG control (bottom) at each genomic feature is shown. TOP2B peaks were defined as common peaks between TOP2B ChIPseq experiments using two different antibodies. **b** Overlap of TOP2B and RAD21 peaks in wild-type thymocytes. **c** Genome browser view of TOP2B, RAD21, CTCF, POLR2A, H3K4me3 (ENCODE) and H3K27ac (ENCODE) signal tracks at a representative genomic region encompassing an active promoter (left, positive for POLR2A, H3K4me3 and H3K27ac) and an insulator (right, negative for POLR2A, H3K4me3 and H3K27ac, and positive for CTCF and RAD21). Sites highlighted in grey show sites of major TOP2B signal accumulation over the region. **d** Heatmaps of TOP2B (this study), POLR2A (ENCODE), RAD21[47] and CTCF (ENCODE) ChIPseq signals at TOP2B peaks (± 5 kb). K-means clusters based on the different signals represented are shown divided by black lines and named as C1 and C2.

TOP2B accumulation were observed at all of these sites, and were particularly enriched at the V(D)J-initiating J segments. As expected, TOP2B coincided with the position of the cohesin subunit RAD21, and, in line with the genome-wide observations, also with the strongest peaks of RAG1 and RAG2. Furthermore, the TOP2B-cohesin sites concurred with a very strong ENDseq signal that was further enhanced in *Atm*$^{-/-}$ thymocytes. Interestingly however, DSB accumulation in these regions was completely dependent on RAG, and did not fully coincide with

TOP2B-cohesin peaks, but was displaced a few kbs from TOP2B sites (Supplementary Fig. 6). This demonstrates that direct RAG cleavage, rather than aberrant TOP2B activity, is the main source of DSBs at *Tcr* and *Igh* loci in thymocytes, but suggests a strong association between TOP2B-cohesin function and sites undergoing active V(D)J recombination. In support of this, TOP2B accumulation in these regions was specific to thymocytes when compared to peaks called in MEFs (Supplementary Fig. 4b, c). ChIPseq of TOP2A also evidenced that TOP2 enrichment was

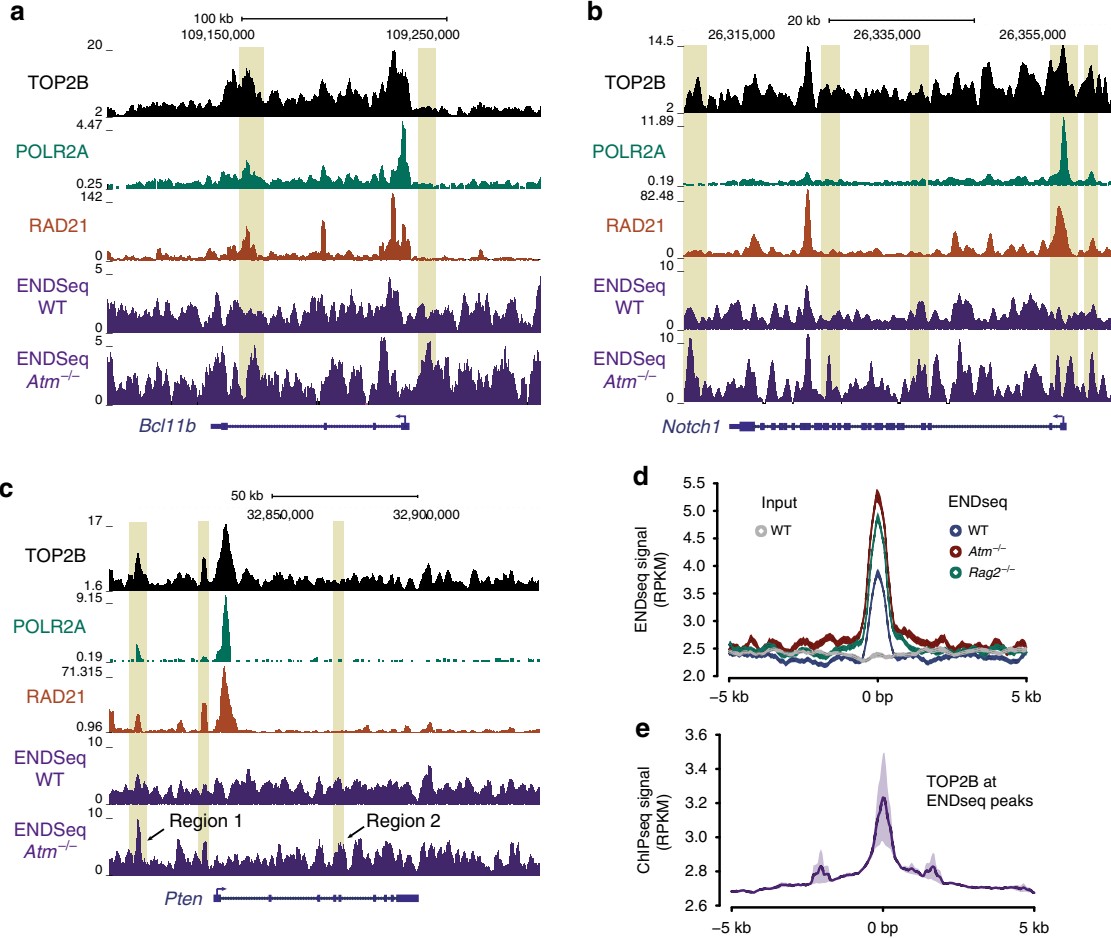

**Fig. 5 TOP2B colocalizes with endogenous DSB accumulation in thymocytes. a–c** Genome browser view of TOP2B, POLR2A, RAD21, and ENDseq (DSBs) signal tracks in wild-type and $Atm^{-/-}$ primary thymocytes, as indicated, at $Bcl11b$ (**a**), $Notch1$ (**b**) and $Pten$ (**c**) loci. $Atm^{-/-}$-specific ENDseq peaks, defined as those not called in wild-type thymocytes, are highlighted in yellow. Two regions of interest in $Pten$ are indicated as region 1 and region 2. **d** Global profile of ENDseq signal at TOP2B peaks in wild-type, $Atm^{-/-}$ and $Rag2^{-/-}$ primary thymocytes. Control input signal is also shown. **e** Global profile of TOP2B signal at ENDseq peaks in wild-type primary thymocytes. Set of ENDseq peaks were determined as the merge between those detected in any of the analysed genetic conditions.

preferential for the TOP2B isoform, both at V(D)J recombination sites and other genes related with the observed oncogenic translocations (Supplementary Fig. 4). Furthermore, when END-seq signal at $Tcrb$ was analysed in detail, there were two peaks that became apparent when the predominant signal derived from RAG cleavage was lost in $Rag2^{-/-}$ thymocytes (Fig. 7c; ENDseq $Rag2^{-/-}$ autoscaled). This relocation of ENDseq signal from major sites of RAG cleavage to additional less prominent sites may be responsible for the apparent increase in DSBs observed at TOP2B peaks in $Rag2^{-/-}$ background (Figs. 5a and 6d). In any case, similarly to what occurred on a genome-wide scale and at other particular locations (Fig. 5), these regions of DSB accumulation were perfectly coincident with the two major sites of TOP2B and RAD21 binding, further supporting an association between TOP2B function and RAG-independent DNA breakage. Nevertheless, despite a similar localized accumulation of TOP2B-cohesin, RAG-independent ENDseq signal was below detection levels at other $Tcr$ and $Igh$ loci (Fig. 7a, b; Supplementary Fig. 5), indicating some degree of variability in the incidence and/or detection of TOP2B-mediated DSBs at these regions.

**$Tdp2^{-/-}$ $Atm^{-/-}$ thymocytes accumulate endogenous TOP2 DSBs.** To directly confirm aberrant TOP2B activity as a relevant

source of DSB accumulation in $Tdp2^{-/-}$ $Atm^{-/-}$ thymocytes, we used a technique recently developed in the laboratory, ICE-IP, in which TOP2 covalently attached to DNA is immunoprecipitated, and the abundance of the genomic region of interest is measured by qPCR (see the "Methods" section and Supplementary Fig. 7 for a detailed description). This allows locus-specific detection of TOP2ccs and TOP2-induced DSBs in which TOP2 has not been fully degraded or removed. In order to be able to detect endogenous lesions, we included a step of nested amplification. We first concentrated on two regions of $Tcrb$ in which RAG-independent ENDseq peaks were detected (region 1 and region 2 in Fig. 7c). Interestingly, TOP2B ICE-IP in freshly isolated thymocytes showed a very clear enrichment of covalently attached TOP2B, exclusively, or at least only significantly, in $Tdp2^{-/-}Atm^{-/-}$ thymocytes, with a >10-fold increase compared to wild-type animals (Fig. 7d). We then tested two regions of $Pten$ with $Atm^{-/-}$-specific ENDseq signal, one that coincided with a strong TOP2B signal and one that did not (region 1 and region 2 in Fig. 5c, respectively). As expected, covalently attached TOP2B accumulated to significant levels only in region 1 of $Pten$ in $Tdp2^{-/-}Atm^{-/-}$ thymocytes, and neither in region 2, nor in wild-type or single-mutant cells. These results strongly support the idea that TDP2 and ATM independently operate to prevent the endogenous accumulation of TOP2B-blocked DSBs, both

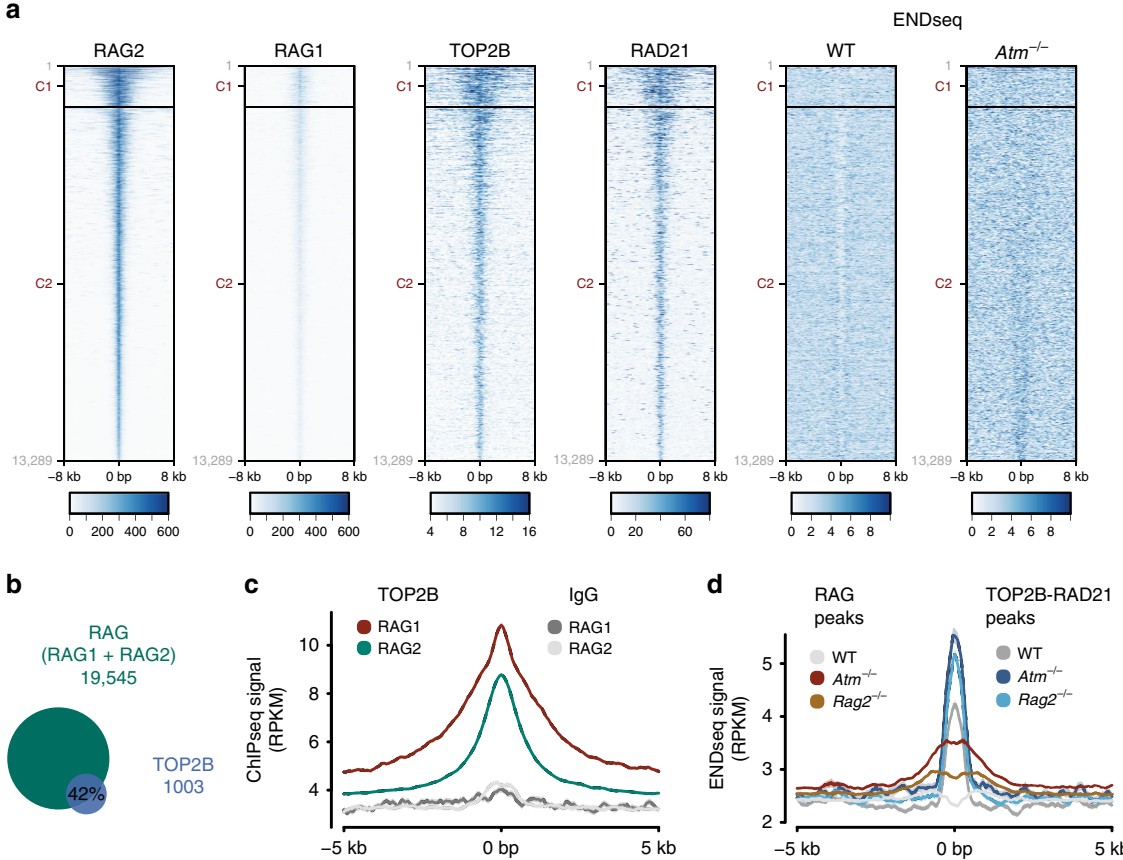

**Fig. 6 TOP2B and cohesin colocalize with RAG genome-wide. a** Heatmaps of RAG2[40], RAG1[40], TOP2B, RAD21 and ENDseq (DSBs) signal at RAG1 and RAG2 peaks (merged) in wild-type and $Atm^{-/-}$ primary thymocytes, as indicated. K-means clusters based on RAG1 and RAG2 signal are shown as C1 and C2. **b** Overlap of TOP2B and RAG peaks. Peaks of the RAG complex were defined as a merge of RAG1 and RAG2 proteins. **c** Global profile of TOP2B signal at RAG1 or RAG2 peaks. Control IgG signal is also shown. **d** Global profile of ENDseq signal at either RAG (merge of RAG1 and RAG2 peaks) or TOP2B-RAD21 (overlap between TOP2B and RAD21) peaks in wild-type, $Atm^{-/-}$ and $Rag2^{-/-}$ primary thymocytes.

globally at particular regions throughout the genome, and specifically at sites undergoing V(D)J recombination.

Finally, in order to reinforce the idea that the rearrangements found in thymic lymphomas could be related to 3D-genome folding and the functions of TOP2B in this process, available Hi–C maps on wild-type thymocytes [41] were integrated with hotspots of $Atm^{-/-}$-linked genomic instability. As a matter of fact, V(D)J recombination sites and other unstable sites were clearly associated with regions of strong long-range interactions (in the order of megabases), apparent as borders of squares in the contact density map (Supplementary Fig. 8), and that could be associated to topologically associating domains (TADs) borders. Interestingly, these regions have been shown to be particularly prone to breakage and instability in general [42,43], and to the accumulation of etoposide-induced DSBs in particular[37], which has led to the idea of TOP2 activity at these functional elements of genome organization as a potential major driver of oncogenic translocations[28,29].

## Discussion

In this study, we show that the absence of TDP2 function significantly aggravates the predisposition of $Atm^{-/-}$ mice to develop thymic malignancies (Fig. 2). Furthermore, TOP2B genome-wide binding correlates with sites of endogenous DSB accumulation (Fig. 5), and endogenous TOP2B-blocked lesions indeed accumulate in $Tdp2^{-/-}Atm^{-/-}$ thymocytes (Fig. 7). These observations strongly support TOP2B-mediated DSBs as

potential drivers of tumour development, at least in the particular setting of A-T-linked cancer predisposition, establishing a causal connection between misrepair of endogenous TOP2 lesions and tumorigenesis. In this sense, the implications of these results reach beyond our understanding of A-T cancer predisposition, since ATM is a well-established tumour suppressor, very frequently mutated in in many tumour types, and lymphoid malignancies in particular[44].

Interestingly, thymic tumours in $Tdp2^{-/-}Atm^{-/-}$ mice display identical characteristics to those of $Atm^{-/-}$ animals, with a very aggressive form of T-ALL associated to recurrent clonal genome rearrangements enriched at loci undergoing V(D)J recombination (Figs. 2–3; Supplementary Fig. 2–3 and Supplementary Table 1). This is, in principle, difficult to reconcile with the incidence of thymic malignancies being increased by loss of TDP2, an enzyme that is highly specialized in TOP2-mediated DSBs, and therefore completely unnecessary for the repair of hairpin DNA ends produced by RAG. Nevertheless, one must bear in mind that translocations require two DSBs, so that TOP2-mediated lesions can contribute to the oncogenic events by providing partners for rearrangements that also engage Tcr or Ig loci, being the co-occurrence of TOP2- and RAG-mediated DNA breaks, and their stabilization and misrepair upon ATM loss, and not RAG-induced breaks per se, what drives ATM-deficient thymic lymphoma predisposition. This would occur in two different scenarios (Fig. 8).

First, we consider TOP2-induced DSBs appearing independently of V(D)J recombination (Fig. 8, top). In this situation,

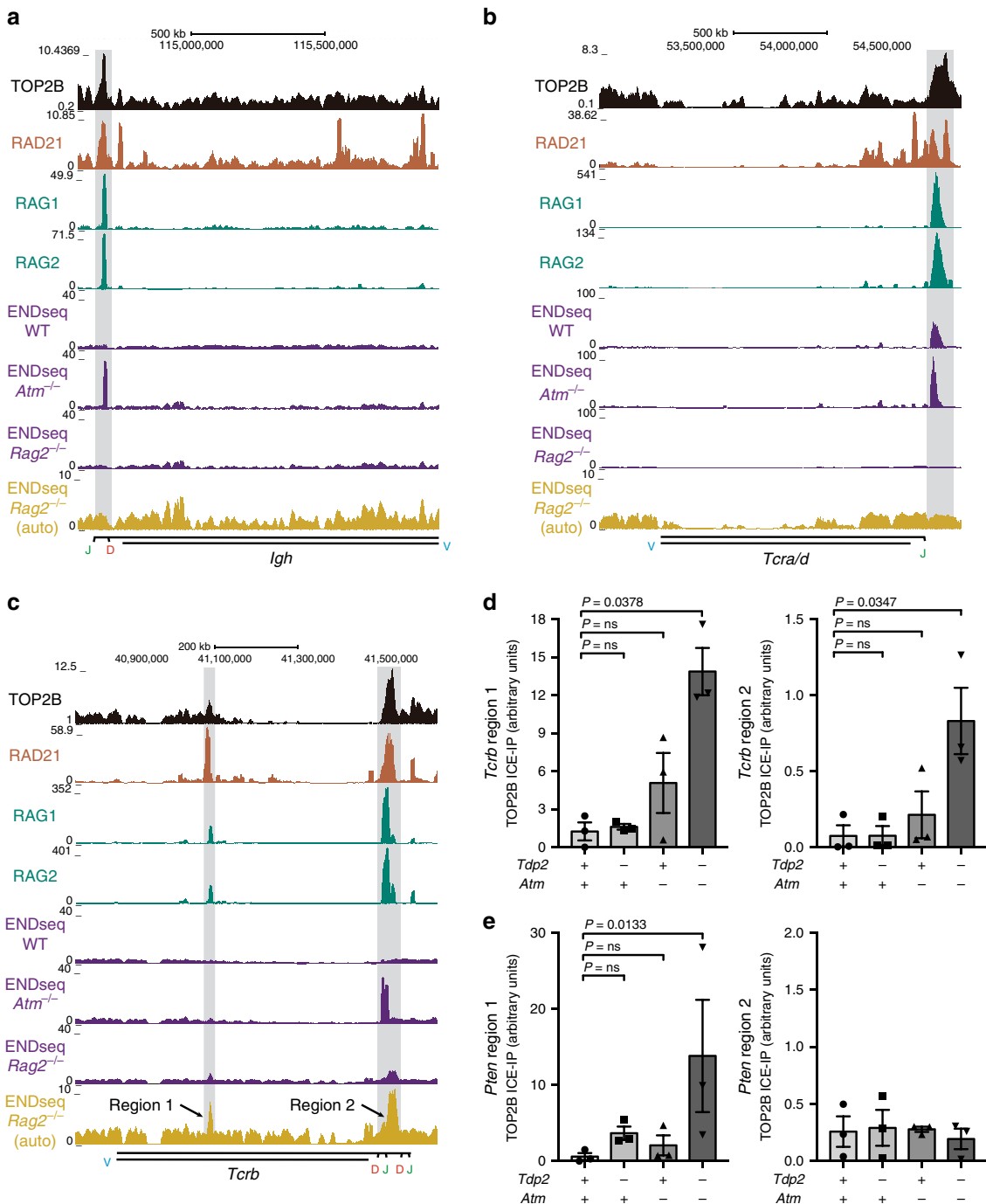

**Fig. 7 TOP2B is enriched at V(D)J-active regions colocalizing with RAG and endogenous DSB accumulation. a–c** Genome browser view of TOP2B, RAD21, RAG1, RAG2 and ENDseq signal tracks in wild-type, $Atm^{-/-}$ and $Rag2^{-/-}$ primary thymocytes, as indicated, at $IgH$ (**a**), $Tcra/d$ (**b**) and $Tcrb$ (**c**) loci. TOP2B enriched regions are highlighted in grey. Regions of V, D or J segments are indicated. ENDseq signal in $Rag2^{-/-}$ thymocytes is additionally shown with a smaller scale to appreciate sites of minor DSB accumulation (yellow, auto). **d, e** Endogenous accumulation of TOP2Bccs and TOP2-mediated DSBs, as measured with ICE-IP, at $Tcrb$ (**d**) and $Pten$ (**e**) loci in $Tdp2^{+/+} Atm^{+/+}$, $Tdp2^{-/-} Atm^{+/+}$, $Tdp2^{+/+} Atm^{-/-}$ and $Tdp2^{-/-} Atm^{-/-}$ freshly isolated thymocytes from three independent mice ($n = 3$). Regions 1 and 2 for $Tcrb$ and $Pten$ are indicated in Fig. 7c and 5c, respectively. Mean ± SEM and statistical significance by one-way ANOVA (non-parametric) with Dunn's post-test is shown ($F = 9$ for region 1 of $Pten$ and $F = 5{,}8$ and $F = 7$ for region 1 and region 2 of $Tcrb$).

aberrant TOP2 activity, which accidentally occurs throughout the genome as a consequence of genome organization or the removal of transcription-associated supercoiling[27,37], results in the formation of DSBs that are normally repaired very efficiently by the action of either TDP2- or ATM-dependent repair pathways[33]. Furthermore, the checkpoint functions of ATM trigger cell cycle arrest and/or apoptosis if these breaks remain unresolved. A

combined failure to properly repair and signal TOP2-mediated DSBs, which can occur accidentally, but is greatly enhanced in the $Tdp2^{-/-} Atm^{-/-}$ mutant, leads to the persistence and propagation of these lesions, as has been shown to occur with RAG-induced DSBs upon single ATM deficiency[15]. This substantially increases the probability of at least two DSBs concurring in time during thymocyte development, providing ideal conditions for

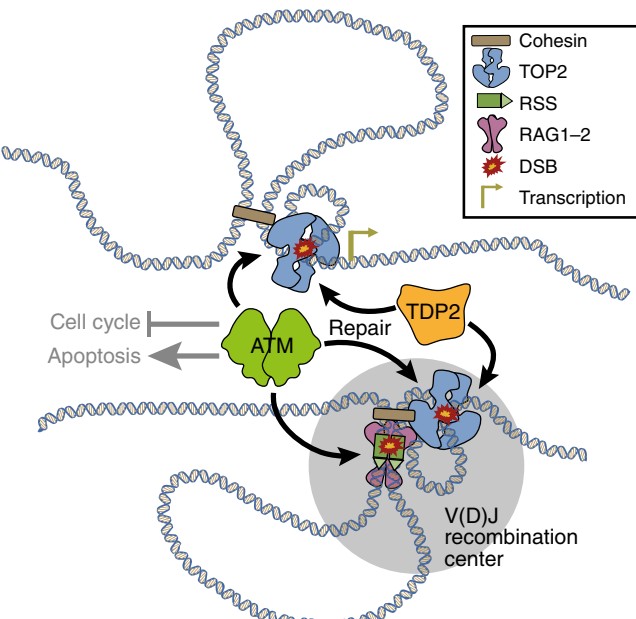

**Fig. 8 Model to explain aberrant TOP2 activity as a driver of ATM-deficient thymic malignancies.** TOP2 activity associated with genome organization and transcription throughout the genome can accidentally result in DSBs (top). Additionally, DSBs can arise as a consequence of TOP2 activity derived from genomic reorganization associated to V(D)J recombination, and in particular, the mechanism of RAG scanning to find its RSS targets (bottom). These DSBs concur with RAG-mediated DSBs, increasing the probability of oncogenic translocations to occur. Efficient repair of TOP2-induced DSBs mediated by either TDP2 or ATM-dependent pathways, together with the checkpoint and apoptotic functions of ATM, strongly limit the oncogenic potential of these lesions.

the generation of chromosomal translocations that can drive oncogenesis. As a matter of fact, irradiation of $Atm^{-/-}$ B-cells has been shown to increase translocations involving the *Igh* locus[15], strongly supporting the idea that additional sources of DSBs can enhance translocation frequency by providing partners for rearrangements with loci undergoing V(D)J recombination. Nevertheless, it is clear that, as previously suggested, off-target RAG cleavage can also constitute a source of additional DSBs during thymocyte development[40]. In this context, the striking genome-wide correlation between TOP2B and RAG binding (Fig. 6) complicates to unambiguously establish their individual contribution to endogenous DNA breakage. The fact of ENDseq signal being quantitatively more relevant at TOP2B than at RAG2 peaks, however, strongly vouches for aberrant TOP2 activity as a major source of endogenous DSB occurrence genome wide.

Second, we consider a scenario in which TOP2-induced DSBs appear associated to V(D)J recombination (Fig. 8, bottom). The chromatin movements and changes in the three-dimensional conformation that necessarily accompany the process of V(D)J recombination result in topological problems that lead to particularly elevated levels of TOP2 activity at these regions, and therefore an increased probability of DSBs to occur. These TOP2-mediated DSBs concur both in time and space with RAG cleavage, constituting a particularly challenging and genome-threatening situation. This possibility is particularly appealing in the context of the recent RAG-scanning models, which invoke cohesin-mediated loop extrusion to explain how RAG finds and pairs RSSs[23]. Once RAG is loaded to form a V(D)J recombination centre, chromatin threading through the cohesin ring would allow RAG to linearly scan the extruded region and find a second RSS for pairing and

cleavage. As has been proposed for general loop extrusion, TOP2 would facilitate RAG scanning by removing topological problems that are encountered by the extruding complex, but, as mentioned above, could result in the formation of DSBs that compromise the integrity of these regions[37]. Thus, the strong thymocyte-specific co-localization of TOP2B-cohesin with RAG at sites actively undergoing V(D)J recombination does not only provide us with a potential additional source of instability at these sites, but can actually be taken as an important support for a loop-extrusion model of RAG scanning.

In summary, we provide a causal link between endogenously occurring TOP2-induced DSBs and cancer development, putting forward these lesions as important contributors to T-cell malignancies related to ATM loss, and opening the door for their potential involvement in other conditions and cancer types. Furthermore, we present evidence for the involvement of TOP2B in resolving topological problems derived from the chromatin movements and changes in genome conformation required for V(D)J recombination, providing an additional source of chromosome fragility at these regions beyond direct RAG cleavage, and supporting a model of loop extrusion-mediated RAG scanning for target RSS pairing.

## Methods

**Animal maintenance.** All animal procedures were performed in accordance with European Union legislation and with the approval of the Ethical Committee for Animal Experimentation of CABIMER and validation by the Regional Government of Andalusia (Consejería de Agricultura, Ganadería, Pesca y Desarrollo Sostenible de la Junta de Andalucía). Double heterozygotes $Tdp2^{+/-}Atm^{+/-}$ and $Tdp2^{+/-}Trp53^{+/-}$ were obtained by the crossing $Tdp2^{+/-}$[32] with $Atm^{+/-}$[4] or $Trp53^{+/-}$[45]. The colonies were maintained by crossing double-heterozygotes and littermates were used for experiments. Animals were housed in isolated cages with controlled ventilation trough HEPA-filters, kept under standard housing conditions (21 ± 1 °C with a photoperiod of 12:12 h), and manipulated in flow cabins. Sterile food pellets and water were available *ad libitum*. Mice were genotyped using Phire Animal Tissue Direct PCR Kit (Thermo) following manufacturer instructions.

**In vivo etoposide sensitivity.** At 8 weeks of age, mice underwent intraperitoneal injection with 3 μl/g of body weight of either DMSO (vehicle control) or etoposide at 25 mg/ml in DMSO for a final dose of 25 mg/kg. Weight and general health status were monitored daily from the day of injection (inclusive). Six days post-treatment mice were sacrificed by cervical dislocation and dissected for histo-pathological analysis. For this, organs were fixed in 4% paraformaldehyde for 2 days, embedded in paraffin, cut in 6 μm slices by microtome, stained with Haematoxylin-Eosin and visualized under the microscope.

**Lifespan analysis.** Minimum 20 mice per experimental condition were included in the analysis. Weight and general health status were monitored weekly. Animals were sacrificed by cervical dislocation and dissected for histopathological analysis in the case of showing a 20% loss of the maximum weight, the presence of a detectable tumour or signs of evident pain. Thymic lymphoma was macroscopically identified and confirmed by histopathological analysis. Overall survival and cumulative occurrence of thymic tumours were determined by Kaplan–Meier curves and statistically analysed with Wilcoxon tests.

**Histological analysis.** Cerebella were fixed in 4% paraformaldehyde, embedded in paraffin, and sagittal 50 μm slices were obtained by vibratome. Immunohistochemistry was performed using anti-calbindin primary antibody (CB-38a, Swant) and Biotin-SP AffiniPure Goat Anti-Rabbit secondary antibody (111-065-003, Jackson). Signal was developed using VECTASTAIN Elite ABC HRP Kit (Pk-6100, Vector Laboratories) and DAB (D4418-50SET, Sigma).

For histological analysis of testicles, organs were fixed in 4% paraformaldehyde, embedded in paraffin, cut in 6 μm slices by microtome and stained with Haematoxylin-Eosin.

**Cellular index assay.** Primary MEFs were isolated from littermate embryos at day 13 p.c. and cultured at 37 °C, 5% $CO_2$ and 3% $O_2$ in Dubelcco's Modified Eagle's Medium (DMEM) supplemented with penicillin, streptomycin, 15% FCS and non-essential aminoacids. MEFs were plated in E-plates 16 in duplicates in a concentration of 4000 cells per well and analysed by xCELLigence® RTCA DP (ACEA Biosciences). The instrument measures cellular proliferation by changes in impedance-based signals in real time. Three independent sets of primary MEFs cells were analysed.

**Lymphocyte analysis**. Healthy thymus (4-weeks old) or thymic lymphomas were extracted from mice, disaggregated to single cells in EDTA-buffer (100 mM NaCl, 1 mM $KH_2PO_4$, 3 mM KCl, 10 mM $Na_2HPO_4$, 1 mM EDTA) and immunostained with anti-CD4-FITC and anti-CD8-APC. Samples were fixed 10 min in 4% paraformaldehyde for storage at 4 °C, and subsequently analysed using a BD FACS-Calibur Flow Cytometer (BD Biosciences). Data was processed and analysed using FlowJo (v9, FlowJo, LLC).

**FACS analysis**. B-cell populations were isolated from 5-week old mice femur passing 2 ml of ice-cold EDTA buffer (100 mM NaCl, 1 mM $KH_2PO_4$, 3 mM KCl, 10 mM $Na_2HPO_4$, 1 mM EDTA) through the femur bone marrow using a 25-G needle syringe. Cells were centrifuged 10 seg at 10,000 rpm in a bench centrifuge and resuspended in 500 μl of PBS-5% FBS (blocking, 30 min). After blocking, 1:100 dilution of conjugated antibodies was added to the samples and incubated on ice for another 30 min. Cells were then washed twice with PBS-5%FBS (centrifuge 10 seg, 10000 rpm) and fixed adding 200 μl of 4%PFA-PBS drop by drop to cells resuspended in 20 μl of PBS. After 10 min of incubation, cells were washed with PBS and analysed by FACScalibur flow cytometer (Becton Dickinson). Antibodies: B220-APC (17–0452) and CD43-FITC (11–0431). CD43$^+$ B220$^+$ cells were considered Pro-B-cells, CD43$^-$ B220$^{low}$ Pre-B-cells and CD43$^-$ and B220$^{high}$ immature B-cells. Data was processed and analysed using FlowJo (v9, FlowJo, LLC).

**CGH analysis**. Genomic DNA was purified from thymic tumours and kidney from $Tdp2^{+/+}Atm^{-/-}$ or $Tdp2^{-/-}Atm^{-/-}$ mice using DNeasy Blood and Tissue Kit (Qiagen) following manufacturer recommendations. One microgram of genomic DNA from thymic lymphoma was profiled against 1 μg of matched normal kidney DNA from the same mouse. Processing, labelling and hybridization to Mouse Genome CGH Microarray 2 × 105 K (G4425B-014699, Agilent) was performed at the CBIMER Genomics Facility. Data was processed using the package snapCGH within R (version 3.3.1). Background correction was applied to fluorescence ratios of scanned images using the method 'minimum'. To compare hybridizations, normalization between arrays by the method 'scale' was performed. Finally, data was processed and ordered by the function 'processCGH', and fluorescence ratios were plotted using the R function 'plot' and the package Gviz[46]. Thresholds for copy number alterations were set at log2 = ±0.6 for trisomy or hemizygous deletion.

**Immunofluorescence**. DSBs in primary MEFs were measured by 53BP1 foci. MEFs were grown on coverslips for 1 day in DMEM supplemented with penicillin, streptomycin, 15% FCS and non-essential aminoacids (37 °C, 5% $CO_2$ and 3% $O_2$) and then fixed for 10 min in ice-cold methanol at −20 °C. Cells were permeabilized for 2 min in PBS−0,2% Triton X-100 and blocked for 30 min in PBS−5% BSA. Coverslips were then incubated with the primary antibody (SantaCruz – sc22760) diluted 1:1000 in PBS−1% BSA for 1 h. After three washes of PBS−0,1% Tween 20, cells were incubated 30 min with the AlexaFluor-conjugated secondary antibody diluted 1:1000 in PBS−1% BSA (Jackson ImmunoResearch − 111-545-144) and washed three times with PBS−0,1% Tween 20. Finally, cells were counterstained with DAPI (Sigma) and mounted in Vectashield (Vector Labs). 53bp1 foci were counted manually (double-blind) in 40 cells per condition using ZEISS ApoTome microscope.

**ICE-IP (in vivo complex of enzymes immunoprecipitation)**. Cells from thymus medulla were isolated from 5-week old mice and placed into RPMI 20% FBS media for 3 h in order to collect only thymocytes in suspension. Cells were then collected by centrifugation at $300 \times g$ 5 min and resuspended into 1% (w/v) N-Lauroylsarcosine sodium salt (Sigma-Aldrich, L7414) in TE buffer with complete protease inhibitor cocktail (Roche) for a denaturizing lysis of cells. Lysates were then homogenized using 25G syringe and subjected to DNA precipitation using a CsCl (Applichem-Panreac, A1098) density gradient. DNA was precipitated only with covalently bound proteins adding CsCl to a final concentration of 0,67 g/ml and centrifuging at 57,000 rpm for 20 h at 25 °C using 3.3 ml 13 × 33 polyallomer Optiseal tubes (Beckman Coulter) in a TLN100 rotor (Beckman Coulter).

Forty micrograms of non-crosslinked DNA were digested for 6 h with HindIII-HF (NEB, R3104) and NdeI (NEB, R0111) restriction enzymes at 1 U/μl each one. Samples were then diluted five times in IP buffer (0,1% SDS, 1% TX-100, 2 mM EDTA, 20 mM TrisHCl pH8, 150 mM NaCl) and incubated o/n at 4 °C with 2 μg of TOP2B antibody (NOVUS, NB100-40842) and then with 25 μL of pre-blocked (1 mg/ml BSA) Dynabeads protein A and Dynabeads protein G (ThermoFisher). Beads were then sequentially washed with IP buffer, IP buffer containing 500 mM NaCl and LiCl buffer (0.25 M LiCl, 1% NP40, 1% NaDoc, 20 mM TrisHCl pH8 and 1 mM EDTA). DNA was eluted by 30 min incubation at 30 °C in 100 μL elution buffer (1% SDS, 100 mM NaHCO$_3$) and then treated with 10 μg of Proteinase K (ThermoFisher) for 2 h at 37 °C prior to purification using Sera-Mag Select beads (GE Healthcare). Sites of interest were amplified using GO-TAQ polymerase and purified again using Sera-Mag Select beads before qPCR with nested primers. In order to avoid PCR saturation, the number of cycles (between 10 and 20) in the first PCR reaction were determined for each pair of primers to results in a range of 25–30 Cqs in the subsequent qPCR. Results for each region of interest were normalized against input and an intergenic control region which does not show significant TOP2B binding.

Primers for PCR: Tcrb-Region1 (FW: GGAGACCCAGAACAGAGCAG, RV: ATAAATAGGGCTGGGGATGG), Tcrb-Region2 (FW: CACCTGCCATAGCTC CATCT, RV: CGGTGATAGCTAGAGGCTGAG), Pten-Region 1(FW: ACTGGCA AGCCAAGCTTAAA, RV: GTGGTTGGTTTCCTGCAGTT), Pten-Region 2 (FW: AACTCCGCTGTGAATTTTGG, RV: CTTTGGGAGGACATGCTAGG) and Control Region (FW: AGGAGAGAATGGAGACAAGAGC, RV: GGTCTCTATC ACTGTTCTCATTGG).

Nested primers for qPCR: Tcrb-Region1 (FW: GGAGACCCAGAACAGAGC AG, RV: ATAAATAGGGCTGGGGATGG), Tcrb-Region2 (FW: AGCTCCATC TCCAGGAGTCA, RV: TGAGGTAGAAAGGGCTGCAT), Pten-Region1 (FW: CCCCTCCCACTTCTATTGT, RV: GTGGTTGGTTTCCTGCAGTT), Pten-Region2 (FW: CCGCTGTGAATTTTGGCTAT, RV: GCCTGTGAAACAGTGC TCAA) and Control Region (FW: TCACTGTTCTCATTGGTTGC, RV: CTCAGG AGTGTCAGGGAAGG).

**ChIPseq**. To prepare chromatin, mouse thymus was extracted from 4–6-week old mice, washed twice with cold PBS and mechanically disaggregated into single cells in 1ml EDTA buffer (100 mM NaCl, 1 mM $KH_2PO_4$, 3 mM KCl, 10 mM $Na_2HPO_4$, 1 mM EDTA). Thymocytes were then washed twice and resuspended again in EDTA buffer. Cells were fixed in a final concentration of 1% formaldehyde and incubated at 37 °C for 10 min. Fixation was quenched by adding glycine to a final concentration of 125 mM. Cells were washed twice with cold PBS in the presence of complete protease inhibitor cocktail (Roche) and PMSF. Cell pellet was lysed in two steps using 0.5% NP-40 buffer for nucleus isolation and SDS 1% lysis buffer for nuclear lysis. Sonication was performed using Bioruptor (Diagenode, UCD-200) at high intensity and two cycles of 10 min (30" sonication, 30" pause) and chromatin was clarified by centrifugation ($17,000 \times g$, 10 min, 4 °C).

For IP, 50 μg chromatin and 4 μg antibody (anti-TOP2B NOVUS-NB100–40842, anti-TOP2B SantaCruz-sc13059, anti-TOP2A Abcam-EP1102Y and rabbit IgG SIGMA-I8140 as control) were incubated o/n in IP buffer at 4 °C, and then with 25 μL of pre-blocked (1 mg/ml BSA) Dynabeads protein A and Dynabeads protein G (ThermoFisher). Beads were then sequentially washed with IP buffer, IP buffer containing 500 mM NaCl and LiCl buffer. ChIPmentation was carried out essentially as previously described using Tagment DNA Enzyme provided by the Proteomic Service of CABD (Centro Andaluz de Biología del Desarrollo). DNA was eluted by incubation at 50 °C in 100 μL elution buffer (1% SDS, 100 mM NaHCO$_3$), cross-linking reverted by incubation with 200 mM NaCl and 10 μg of Proteinase K (ThermoFisher) o/n at 65 °C and DNA purified using Qiagen PCR Purification columns. Libraries were amplified for N-1 cycles (being N the optimum Cq determined by qPCR reaction) using NEBNext High-Fidelity Polymerase (M0541, New England Biolabs), purified using Sera-Mag Select Beads (GE Healthcare) and sequenced using Illumina NextSeq 500 and single-end configuration.

**NGS analysis**. Tags were aligned to mouse genome (mm9) using Bowtie (version 1.2.0). Peaks were called using HOMER and filtered by a fold change enrichment over control (IgG ChIPseq) of 10-fold and standard quality parameters for "factor-style" peak calling. When comparing peak colocalization, TOP2B called peaks were extended 1 kb in each direction. For genome tracks, we used bamCoverage (deepTools) to convert aligned reads to signal tracks (bigwig) using RPKM normalization. Visualization was done by UCSC browser and profiles and heatmaps were generated by Seqplots applying k-means algorithm for clustering. Two experiments with two different antibodies against TOP2B were performed in the study and one for TOP2A isoform. In order to increase the confidence of TOP2B peaks, TOP2B peak dataset was based on both experiments and defined as the common peaks called in the two replicates. For genome browser tracks, TOP2B ChIPseq performed using NOVUS Biologicals antibody (NB100-40842) was used.

Genome annotation was obtained from UCSC classification. V, D, J segment annotations were obtained from Ji et al.[39]. All comparisons are performed with datasets derived from thymocytes except RAD21 data, which is derived specifically from CD4/CD8 DP thymocytes. Since DP is the most abundant (>90%) population of thymocytes, we consider this dataset valid for comparison with experiments performed in total thymocytes. RAG1 and RAG2 ChIPseq in thymocytes were obtained from Teng et al. 2015 (SRA: PRJNA285688)[40], thymocyte ENDSeq datasets from Canela et al. 2016 (SRA: PRJNA326246)[38], TOP2B ChIPseq in MEFs from Canela et al. (SRA: PRJNA387544)[37] and RAD21 ChIPseq in CD4$^+$CD8$^+$ DP thymocytes from Loguercio et al. (SRA: PRJNA432324)[47]. POLR2A, H3K4me3, H3K27ac and CTCF thymocyte datasets were obtained from ENCODE (ENCSR000CEA, ENCSR000CCJ, ENCSR000CCH and ENCSR000CDZ respectively). All the raw reads from the mentioned available ChIPseq and ENDseq experiments were processed following the pipeline described above.

**Hi–C analysis**. Hi–C data was obtained from Falk et al.[41] (GEO accession number: GSE111032) and available contact matrixes displayed using HiGlass web-tool (http://higlass.io).

**Software and algorithms**. UCSC Genome Browser – Kent et al. 2002. https://genome.ucsc.edu

HiGlass - Kerpedjiev et al. 2018. https://higlass.io/

Bowtie 1.2.0. – Langmead et al. 2009. https://sourceforge.net/projects/bowtie-bio/files/bowtie

MACS – Zhang et al. 2008. https://pypi.python.org/pypi/MACS
HOMER – Heinz et al. 2010. https://homer.ucsd.edu/homer
SAMTOOLS-1.1 – Li et al. 2009. https://github.com/samtools/samtools
deepTools-2.4.1. – Ramirez et al. 2016. https://deeptools.readthedocs.io
R – R Development Core Team, 2008. https://www.r-projects.org
SeqPlots – Stempor et al. 2016. https://github.com/Przemol/seqplots
GraphPad Prism v6. for statistical analysis. https://graphpad.com
FastQ Toolkit - http://hannonlab.cshl.edu/fastx_toolkit/
Trimmomatic - Bolger et al. 2014. https://doi.org/10.1093/bioinformatics/btu170

BD CellQuest pro and FlowJo v9. for FACS analysis.

**Reporting summary**. Further information on research design is available in the Nature Research Reporting Summary linked to this article.

## Data availability

Next-generation sequencing data are available at the Gene Expression Omnibus (GEO) database under accession number GSE133954 and CGH arrays have been deposited at the GEO database under accession number GSE134054 All the other data supporting the findings of this study are available within the article. Additionally, a source data file for Figs. 1–7 and Supplementary Figs. 1 and 2 is provided.

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

## Acknowledgements

This work has been funded with grants from the Spanish and Andalusian Government (SAF2010-21017, SAF2013-47343-P, SAF2014-55532-R, SAF2017-89619-R, CVI-7948, European Regional Development Fund), and the European Research Council (ERC-CoG-2014-647359); and with individual fellowships for AAQ (Formación Personal Investigador, BES-2011-047351, Ministerio de Ciencia e Innovación), JTB (Formación Profesorado Universitario, FPU15/03656, Ministerio de Educación, Cultura y Deporte) and ASB (Beca Predoctoral AEFAT, Asociación Española Familia Ataxia Telangiectasia). CABIMER is supported by the Andalusian Government. Computational analyses were run on the High Perfomance Computing cluster provided by the Centro Informático Científico de

Andalucía (CICA). We thank the Genomics facility at CABIMER for CGH and NGS experiments, O. Fernández-Capetillo for reagents and A. Aguilera for comments.

## Author contributions

Conceptualization: A.A.Q., J.T.B. and F.C.L.; Experimentation: A.A.Q., J.T.B., I.D.S., A.S.B., R.R.G., C.B.L., C.Q. and L.G.Q.; Data analysis: J.T.B., A.A.Q., P.M.M.G., S.J.G. and F.C.L.; Writing: A.A.Q., J.T.B. and F.C.L.; Funding Acquisition: F.C.L.; Supervision: F.C.L.

## Competing interests

The authors declare no competing interests.
