## [Peer Review File · Nature Communications]

Reviewers' comments:

Reviewer #1 (Remarks to the Author): Expertise in DSB repair

In this manuscript, the Authors generate a *Tdp2*^{-/-} *Atm*^{-/-} double-deficient mouse model to uncover topoisomerase II-induced DSBs as major additional drivers, besides RAG, of the genomic rearrangements that underlie lymphoid malignancies. This work is particularly relevant in the context of the human ATM loss-of-function disorder Ataxia-Telangiectasia (A-T), a neurodegenerative syndrome that predisposes for lymphoid cancers driven by spontaneous chromosomal instability.

This study approaches endogenous, TOP2-mediated DSBs by removing TDP2, an essential factor in the normal TOP2 action cycle. In the background of ATM deficiency, this proves an elegant method of increasing TOP2-DSBs while circumventing the need for treating with a TOP2 poison.

Overall, the work is of high quality, and the manuscript is generally very well written and interesting to read. However, I feel that certain parts, particularly those based on End-seq datasets, require more attention and that more details need to be added to the Manuscript. Furthermore, the conclusions would likely be strengthened, if the Authors expanded the spectrum of genome-wide analyses, by performing additional experiments and/or analyzing more available datasets.

MAJOR REMARKS

1) Rationale and flow of the manuscript.

Although overall the Manuscript is thoughtfully put together and well written, the Introduction but also parts of the Results section should be reshaped to let the distinct topics connect better to each other and to the overall goal, which should be more clearly stated as now there are various open questions in the Introduction. Furthermore:

-- The Authors should describe the action of TOP2 and TDP2 in a more structured way. Attention should be given first to normal TOP2 activity (including that supercoiling builds up around replication and transcription machinery), the cleavage complex, and only then should the Authors focus on the consequences of aberrant TOP2 action in which the DSB is not transient and quickly re-ligated, and poisons that act on this. It is not clear whether TDP2 is involved in normal resealing of the transient DSB as well, or only in DSBs formed through aberrant/abortive action. This should be better clarified, as well as whether TDP2 has any known other functions. Furthermore, it would be useful if the Authors also mentioned in the Introduction the recent work by the Roukos and Nussenzweig labs (Gothe et al. and Canela et al., *Mol Cell* 2019) linking TOP2 poisons to DNA breaks underlying translocations that drive secondary lymphoid malignancies, to further strengthen that TOP2-induced DSBs challenge genome integrity and have been linked to cancers before. The description in the Discussion is much clearer and can maybe be combined into the Introductory text.

-- In the Introduction the Authors state that TDP2 is the only human enzyme with relevant 5'-tyrosyl-DNA-phosphodiesterase activity. In light of the relevance of their double knockout mice model, they should comment on whether the same is true for mouse.

-- The experiments on the p53 and TDP2 double-deficient mice are elegant and valuable, but the rationale behind this choice as well as how the conclusion is reached should be explained better.

-- Starting from line 360, the Authors describe their results of TOP2B and END-seq signal distributions and they describe their finding of accumulation of DSBs at TOP2B peaks as interesting and surprising. In light of recently published work (Gothe et al. and Canela et al., *Mol Cell* 2019), I

believe this overlap should not be considered that surprising as after all, TOP2B is considered a main source of endogenous breaks, albeit mostly transient ones. In line with this, the end of the Abstract suggests that this work opens the door for involvement of TOP2-induced DSBs in other cancer types and disease conditions. I believe this statement should be toned down, because this involvement has already been hypothesized and revealed before.

3) Experiments and results.

-- The individual CGH profiles that are averaged in Figure 3A show a lot of variation between the different animals (Supp. Fig). In addition to the observed similarities that stand out when the profiles are averaged, there is a lot of variation and especially in the scores of deletion between them. Can the Authors comment or speculate on this?

-- Related to the previous comment, the resolution afforded by aCGH is limited and, as the Authors themselves acknowledge, it is very difficult to infer the original breakpoints where DSBs must have occurred in the first place. In this sense, it would be very useful if the Authors applied whole-genome sequencing to characterize in greater depth and resolution the rearrangements in thymic tumors. The Authors would then be able to test the hypothesis that frequently rearranged loci are also DSB hotspots.

-- It is not clear from the experiments and analyses shown how much the rearrangements found in thymic tumors are related to the interplay between 3D genome structure and TOP2 activity. To this end, it would be valuable if the Authors could overlay their rearrangements data to Hi-C maps previously obtained from mouse thymocytes (for example, in Seitan et al., 2013), and explore whether frequently rearranged regions in thymic tumors overlap with specific TAD borders and/or are enriched in particular 3D compartments.

-- In line 372, the Authors write that lesions can be stabilized upon ATM deficiency. It is not clear to me what is meant here: are the lesions (cleavage complexes) stabilized into full DSBs? In lines 84–85 the word stabilizing is used to describe the role of ATM in V(D)J recombination, where it stabilizes the post-cleavage complex and protects the DNA ends against incorrect ligation. Then, when speaking about ATM deficiency and in the context of lesion repair, this word choice is contradictory. The Authors should rewrite this sentence to make it less ambiguous.

-- Can the Authors comment on why the END-seq signal at TOP2B sites increases (rather than staying similar) in the Rag2^{-/-} thymocytes in Fig. 5D?

-- The Authors should explain better in the method section why they use two different antibodies for TOP2B ChIPseq. Now the only place to find this information is in the Figure legend.

-- In the methods section, the Authors should add that the ENCODE datasets they used were generated in mouse thymocytes. Furthermore, datasets from different sources and in some cases different cell types were used. This should be more explicitly mentioned in the manuscript, including the possible implications this may have.

4) Analyses.

-- The Authors should be more explicit in their description of their analyses, particularly with regards to the NGS datasets. How are insulators and enhancers called and defined? How many peaks were called, and in the figures such as 4D, what exactly is a peak and what is not? Is the entire region bound by TOP2B or is everything between the grey regions considered background?

-- Following the previous point, how are enriched regions defined in the END-seq data? For example in Fig. 5B, there are many peaks that, to the untrained eye, seem to be strong and enriched with respect to the WT, but they are not indicated with yellow. Furthermore, in line 364

and Fig. 5E the Authors regard the TOP2B signal at END-seq peaks, but how are peaks defined here, with what cut-off? To make the difference between the WT and the deficient thymocytes profiles more prominent, for example the significant differences should be marked by a different color, indicating differential peaks, or peak-called enriched regions should be more clearly indicated.

-- The Authors should make a note about the variation of y axis ranges of the END-seq data. Comparison between the axes of Figure 7 and Figure 5 and Supp. Fig 5 shows huge differences, and the differences in Figure 5 seem very small compared to those in Supp. Fig 5. Maybe the axes in Fig. 5 can be extended, because the current view does not fit well with the description that 'no substantial END-seq signal was detected' (lines 351 and 352).

-- The Authors should add an intersection of the END-seq data with the observed genome rearrangements in Figure 3 (or all the observed breakpoints in all animals as shown in the Supplementary Figure) to show that indeed, in the ATM^{-/-} compared to the WT, there is an enrichment of breaks in those regions (in a more quantitative manner than with genome browser shots of the most relevant genes).

-- In Figure 5E, the TOP2B signal at END-seq peaks shows 'shoulders' in addition to the major center peak. The Authors should comment on these, perhaps by speculation.

-- Can the Authors quantify the genome-wide co-localization in percentage of peaks that overlap between TOP2B, RAG1, and RAG2, perhaps by extending the Venn diagram of Fig. 4, and refer to it in line 401?

-- How does Fig. 6C relate to Fig. 6A, the patterns of END-seq signal around RAG peaks in the WT and the two mutants seem different (more enriched in case of the ATM mutant) than can be observed in 6A? Also, is there an explanation for why the END-seq signal goes up around RAG peaks when Rag2 is removed?

-- To appreciate the few-kb displacement of DSB peaks in Figure 7, please add a zoom-in with a ruler to make this detail more appreciable.

ADDITIONAL REMARKS

-- The title of the manuscript differs from the title on the Supplementary Information file.

-- In Fig. 2B the y-axis should be labeled Cumulative risk of developing thymic tumor to make the figure more directly comprehensible (same for all axes labeled Cumulative risk). Same holds true for other figures with this graph (e.g. risk of developing cancer in general, risk of developing thymic tumors).

-- In line 87, it is unclear what 'These defects in V(D)J recombination' refers to. In the alinea above this sentence, normal V(D)J recombination is described, including the role of ATM, and how RAG finds RSS sequences. The Authors should describe the defects they refer to in more detail or remove the word these.

-- Supplementary Figure 2A should be accompanied by a table indicating percentages of the respective precursors.

-- In line 187, a reference is made to Supp. Fig. 1D, but it needs to be adjusted to 1C.

-- Line 280 refers to Figure 3C, but there is no such figure in the manuscript I received.

-- All graphs should be given axes titles.

-- Figure 5 Legend does mention (D) twice, this should be correct to (D) and (E).

-- In line 420 the Authors refer to Supp. Fig. 4, stating that TOP2B accumulation in these regions is absent or largely decreased in MEFs. I don't agree with this statement, because in C the reduction is very minimal (nowhere near largely decreased).

-- In line 430 the Authors refer to Supp. Fig. 4B, which should most likely be corrected to 5B.

-- It is not clear to me why the Authors refer to the END-seq columns in Fig. 6A as RAG-dependent END-seq signal? Please check and/or clarify.

-- In the Discussion in line 549, reference is made to Figure 7 to show that TOP2B-RAG co-localization not only occurs at V(D)J regions but also genome-wide. Figure 7 shows no genome-wide co-localization as far as I see, but indeed the authors should add a figure showing that (a point that I have raised further above as well).

Reviewer #2 (Remarks to the Author): Expertise in TOP2B and genome architecture)

The authors report a strong combinatorial effect of loss of the DNA repair enzyme Tdp2 and ATM in the frequency of lymphoid cancers in a mouse model. These tumours are common in the ATM^{-/-} mouse model and in A-T patients and their aetiology is thought to involve inefficient and inaccurate repair of RAG-induced DSBs during antigen and T-cell receptor recombination. Since Tdp2 is required for a major mechanism for repair of TOP2-DNA adducts that may occur spontaneously in the genome, the authors conclude that endogenous TOP2 DNA lesions underlie the combined effect of Tdp2 and Atm loss. They also allude to the fact that part of the mechanism through which ATM leads to elevated tumour risk, especially in Tdp2 nulls could be via modulation of a nucleolytic pathway for TOP2-DNA adduct resolution (although this could be made much clearer). The first part of the paper describes the characteristics of Tdp2^{-/-}, ATM^{-/-} mice, the frequency of thymic tumour formation and genetic instability (CGH analysis) while the second part concerns TOP2B ChIP-seq analysis both globally and at specific loci. For this the authors performed their own TOP2B ChIP-seq in primary mouse thymocytes and interrogated this alongside available published data including End-seq data from Canela et al (2016) and Rag ChIP-seq from (Teng et al 2015). From these analyses the authors conclude that TOP2 occupancy colocalises to an extent with sites of spontaneous DNA breaks (End-seq) and with sites of RAG binding throughout the genome. In addition, the authors demonstrate that TOP2B is enriched at VDJ recombination regions in proximity to sites of RAD21 and RAG binding. At these sites there is a strong peak of endogenous DNA breaks (End-seq) which is RAG dependent. The authors conclude that their findings demonstrate a strong causal relationship between what they refer to as "accidental TO2B-induced DSBs" and cancer development.

Overall, I feel that the authors have produced some interesting and important findings relating to DNA damage and the sources of genome instability in the absence of clastogenic insult. The experiments appear to be well performed and the work is timely and significant. but there are number of major and minor issues that need addressing.

Major Issues

1. Page 8/ Fig1 There is a lot made in this paper about increased accumulation of spontaneous TOP2 lesions and thus endogenous DNA damage in the absence of both ATM and Tdp2, but there is no direct evidence for this presented and this is a serious weakness. Elevated rates of "accidental" TOP2-mediated damage could perhaps be detected as increased numbers of

“background” gH2AX foci in MEF cells, or ideally through comparing End-seq in WT, ATM^{-/-}, Tdp2^{-/-} cells and cells null for both.

2. Page 14, Fig 5. There is some significant over-interpretation here. Most seriously regarding the End-Seq data. It is stated that “the main peak of TOP2B-RAD21 binding at Notch was perfectly aligned with a region of increased End-seq sequence” – the End-seq data in this region is “a choppy sea”, i.e. a continuous series of small peaks, possibly representing the noise in the system, and similar to the End-seq pattern for the other genes shown. The chances of one of these peaks lining up with a TOP2 peak by chance is clearly high, and so this data cannot be used as evidence of endogenous DNA damage concentrated at these TOP2B peaks. The same problem applies to the claim that in ATM^{-/-} cells peaks appear that coincide with the TOP2/RAD21 signal. Similarly, there are thousands of robust TOP2B peaks across the genome, about a third in promoters, so there is nothing special about the pattern of TOP2B distribution at the Bcl11b, Pten or Notch1, so it is not clear how this is connected with the genome instability evidenced by the CGH data.

3. The source of the existing End-Seq and ChIP-seq data sets that were interrogated along with the TOP2B ChIP-seq data originating in this study must be acknowledged in the legends to Figs 4 and 5 at least, and possible also noted in the main Results text.

4. Line 39, Fig 6: The possibility that the apparent overlap between TOP2/RAD21 peaks and RAG2 peaks could arise because they are both overrepresented in H3K4Me3 rich promoter regions should be considered.

5. The discussion is much too long and should be considerably shortened.

6. More ChIP-seq/NGS details are required about the number of replicate samples, number of aligned reads etc and where the new data reported here (thymocyte Chip-seq) is deposited (although the latter may be something that is inserted after acceptance)

Minor Issues

1. Main manuscript and Supplemental material have different titles.

2. Line 81: Remove “being”

3. Line 84: Swap “thus” and “protecting”

4. Line 102-107: Confusing sentence, needs rewording

5. Line 108: for clarity I think “in ATM^{-/-}” should be inserted between “in” and “mice”.

6. Line 118: RAG scanning is introduced here, but with no explanation of what it is, I think a brief description is required here for the benefit of the likely audience

7. Line 120: Is Ref 12 the correct one here, shouldn't it be Ref 13?

8. Line 122: Should the word “fully” be inserted between “not” and “suppress”

9. Line 130: replace “on” with “in”.

10. Page 6: Should also elude to nucleolytic processing pathway/s for resolving and repairing TOP2-DNA adducts involving MRN, CtIP and/or Artemis etc. This is an odd omission given that ATM could modulate this.

11. Page 6 bottom: Should give a reference for dietary factors such as flavonoids that could contribute to TOP2-DNA lesions.

12. Page 7 top: Given that there are frequent references to “accidental TOP2-induced DNA damage” consider mentioning here that existing endogenous DNA damage such as the presence of nicks or abasic sites may poison TOP2 in situ.

13. Page 155: Refs 41, but not Ref 42 shows this (42 just cites 41)

14. Page 7, 2nd para, see point 10 above, this is a bit lacking on mechanistically how lack of ATM could lead to an accumulation of TOP2-DNA adducts and could also refer to work showing that ATLD cells exhibit elevated basal TOP2 complexes, consistent with a role for MRE11 in processing TOP2-DNA complexes.

15. Line 180: should be “neither did it aggravate”.

16. Fig 1B, y axis mislabelled, presumably it should be % of original weight rather than Weight loss (%).

17. Line 203-204: Confusingly worded

18. Line 216, Replace “in” with “over”

19. Line 282: should be ref 26??

20. Line 323: CTCF – typo

21. Fig 4. This could be moved into the Supplementary material, the data in this Fig is really just validating the thymocyte TOP2B ChIP-seq and doesn't show anything new or surprising.

22. Fig 4D, why was Cxcr4 in particular chosen?

23. Fig 5: There are two Ds and no E in the Figure legend

24. Fig 6C: End-seq signal at TOP2B/RAD21 peaks appears to be higher in Rag2^{-/-} cells than in WT, this is not discussed, but should be.

25. Line 434-435; The term "accidental TOP2 activity" used here and elsewhere is problematic. Presumably TOP2 has some real function at these sites, probably resolving supercoiling issues, so its activity is not accidental. I assume what is meant is incidental or unscheduled DNA damage resulting from (presumably) rare events where TOP2 fails to complete its reaction cycle leaving a stalled TOP2-DNA complex that is resolved either by the proteasome-Tdp2 or nucleolytic pathways. These instances need to be corrected and written more precisely.

Reviewer #3 (Remarks to the Author): Expertise in ATM and in vivo

This study by Alvarez-Quillon et al seeks to explore the causal relationship between TOP2-induced double strand breaks and cancer development focusing the attention in genome rearrangements in the absence of ATM kinase.

The elucidation of Topoisomerase functions and mechanisms in cellular systems deficient for DNA damage repair is relevant and of great interest for understanding the onset and progression of tumorigenesis.

The authors obtained a double knockout mouse model lacking of Atm kinase, the key regulator of DSBs repair and of TDP2 enzyme that unblocks TOP2-induced DSBs allowing the DNA repair. They show that double knockout mice have a phenotype with similarity with the Atm knockout mice in term of developmental defects and thymic genomic rearrangements.

The authors claim that compared to Atm knockout mice, the double knockouts display increased thymoma predisposition and are more sensitive to etoposide exposure. They have previously shown that etoposide stabilizes TOP2 cleavage complexes in double Atm and TDP2 knockout MEFs. In support to their work the authors find enrichment of TOP2B in spontaneous DSBs of thymocytes with highest accumulation at the RAG sites and V(D)J-active regions.

Overall, the data in the present manuscript are insufficient to shed significant light on the topic and to support the conclusions of the authors, preventing the acceptance for publication.

The two major criticisms of the paper regard the comparative analysis of tumorigenesis between the double knockout mice and the Atm knockout model and the study limited to TOP2B isoform.

Comments below are provided as suggestions to improve this study.

Major points:

1. Other authors have shown that etoposide affects the survival of MEF cells obtained from Atm knockout embryos also at low concentration (Yamamoto K. et al., eLife 2016). A dose - response curve and quantification of etoposide damage should be done in double KO and Atm KO mice.
2. The double knockout mice need further characterization to reach conclusions on their phenotype during development and cancer predisposition. Indeed B cells maturation and class switching recombination and meiotic differentiation must be evaluated in double KO compared to Atm KO mice.
3. Figure 1 A and 1B need clarification to make conclusions on thymic lymphoma predisposition of double knockout mice vs Atm KO mice. The relative Kaplan-Meier survival curves do not appear significant. In the figure legend should be reported how statistics has been performed. Statistic calculated only in the mean value of survival is incorrect. How was detected thymoma? Which is

the percentage of double KO and Atm KO mice that develop thymoma at different ages? (ie: at 3, 6, and 12 months). How the authors evaluated the cumulative risk? A description must be included in the methods section. Re-evaluation of these analyses is required also to state that latency of malignancy do not change in the two mouse models.

4. The extensive genomic analysis done for TOP2B in wild-type thymocytes must be implemented with the analysis of TOP2A isomerase that is expressed in proliferating cells of the thymus that also undergo genomic rearrangements and it is associated with cancer predisposition.

Minor points:

1. A comment on the different response observed in the small intestine of Atm KO and TDP2 KO mice exposed to etoposide should be inserted in the results.
2. Atm KO mice in standard housing conditions die, mainly for thymoma, between 4 and 6 months of age depending from the animal facilities, some of them reaching one year of age. The long life span (more than 1 year) reached by several AtmKO and double KO mice in the house condition described in the methods deserves attention and consideration to drive conclusions of in the study.
3. Authors should consider the use of Atm inhibitors in vivo to understand the structural Atm function in the repair of TOP2-induced DSBs.
4. Check the correspondence of the figures in the text. i.e. Fig 3b, Suppl fig 1C.
5. Check P value in figure legend 1B.

We would like to thank the referees for their time and their very useful comments, which have undoubtedly improved and strengthened the manuscript, both experimentally and conceptually. We provide below a point-by-point response to each of the comments, hoping that the manuscript can now be considered suitable for publication.

Reviewers' comments:

Reviewer #1 (Remarks to the Author): Expertise in DSB repair

In this manuscript, the Authors generate a *Tdp2*^{-/-} *Atm*^{-/-} double-deficient mouse model to uncover topoisomerase II-induced DSBs as major additional drivers, besides RAG, of the genomic rearrangements that underlie lymphoid malignancies. This work is particularly relevant in the context of the human ATM loss-of-function disorder Ataxia-Telangiectasia (A-T), a neurodegenerative syndrome that predisposes for lymphoid cancers driven by spontaneous chromosomal instability.

This study approaches endogenous, TOP2-mediated DSBs by removing TDP2, an essential factor in the normal TOP2 action cycle. In the background of ATM deficiency, this proves an elegant method of increasing TOP2-DSBs while circumventing the need for treating with a TOP2 poison.

Overall, the work is of high quality, and the manuscript is generally very well written and interesting to read. However, I feel that certain parts, particularly those based on End-seq datasets, require more attention and that more details need to be added to the Manuscript. Furthermore, the conclusions would likely be strengthened, if the Authors expanded the spectrum of genome-wide analyses, by performing additional experiments and/or analyzing more available datasets.

MAJOR REMARKS

1) Rationale and flow of the manuscript.

Although overall the Manuscript is thoughtfully put together and well written, the Introduction but also parts of the Results section should be reshaped to let the distinct topics connect better to each other and to the overall goal, which should be more clearly stated as now there are various open questions in the Introduction.

We have made an effort to streamline the Introduction. We hope now the different topics are clearer and better connected.

Furthermore:

-- The Authors should describe the action of TOP2 and TDP2 in a more structured

way. Attention should be given first to normal TOP2 activity (including that supercoiling builds up around replication and transcription machinery), the cleavage complex, and only then should the Authors focus on the consequences of aberrant TOP2 action in which the DSB is not transient and quickly re-ligated, and poisons that act on this. It is not clear whether TDP2 is involved in normal resealing of the transient DSB as well, or only in DSBs formed through aberrant/abortive action. This should be better clarified, as well as whether TDP2 has any known other functions. Furthermore, it would be useful if the Authors also mentioned in the Introduction the recent work by the Roukos and Nussenzweig labs (Gothe et al. and Canela et al., Mol Cell 2019) linking TOP2 poisons to DNA breaks underlying translocations that drive secondary lymphoid malignancies, to further strengthen that TOP2-induced DSBs challenge genome integrity and have been linked to cancers before. The description in the Discussion is much clearer and can maybe be combined into the Introductory text.

Performed as requested (lines 108-131).

-- In the Introduction the Authors state that TDP2 is the only human enzyme with relevant 5'-tyrosyl-DNA-phosphodiesterase activity. In light of the relevance of their double knockout mice model, they should comment on whether the same is true for mouse.

The same is indeed true for mouse. Changed accordingly (line 134-135).

-- The experiments on the p53 and TDP2 double-deficient mice are elegant and valuable, but the rationale behind this choice as well as how the conclusion is reached should be explained better.

Performed as requested (lines 244-260).

-- Starting from line 360, the Authors describe their results of TOP2B and END-seq signal distributions and they describe their finding of accumulation of DSBs at TOP2B peaks as interesting and surprising. In light of recently published work (Gothe et al. and Canela et al., Mol Cell 2019), I believe this overlap should not be considered that surprising as after all, TOP2B is considered a main source of endogenous breaks, albeit mostly transient ones. In line with this, the end of the Abstract suggests that this work opens the door for involvement of TOP2-induced DSBs in other cancer types and disease conditions. I believe this statement should be toned down, because this involvement has already been hypothesized and revealed before.

We have toned down these statements as requested (lines 45-48). Nevertheless, we would like to point out that, although we agree that TOP2 has been indeed hypothesized to be a main source of endogenous DSBs, previous studies have been based on lesions induced with etoposide. We have made an effort throughout

the manuscript to clarify this, and to place the novelty of our findings in context.

3) Experiments and results.

-- The individual CGH profiles that are averaged in Figure 3A show a lot of variation between the different animals (Supp. Fig). In addition to the observed similarities that stand out when the profiles are averaged, there is a lot of variation and especially in the scores of deletion between them. Can the Authors comment or speculate on this?

Yes, this is indeed the case, and is similar to what has been reported for the *Atm*^{-/-} single mutant. We understand that this is due to the fact that the genome rearrangements associated with tumour onset and progression have an important degree of stochasticity. We have clarified this in the text (line 298-301).

-- Related to the previous comment, the resolution afforded by a CGH is limited and, as the Authors themselves acknowledge, it is very difficult to infer the original breakpoints where DSBs must have occurred in the first place. In this sense, it would be very useful if the Authors applied whole-genome sequencing to characterize in greater depth and resolution the rearrangements in thymic tumors. The Authors would then be able to test the hypothesis that frequently rearranged loci are also DSB hotspots.

We completely agree that the resolution of CGH is limited, and that being able to precisely map breakpoints by whole-genome sequencing would be useful. We do not think, however, that the impact that these results would have on strengthening the main conclusions of our study justify the level of effort required for these experiments, specially considering the number of tumour samples that would need to be sequenced and analysed to obtain meaningful results.

Whole-genome sequencing of tumours in A-T patients, and a comparative analysis with major sites of TOP2 binding and spontaneous TOP2-DSBs would be an important avenue for future research.

-- It is not clear from the experiments and analyses shown how much the rearrangements found in thymic tumors are related to the interplay between 3D genome structure and TOP2 activity. To this end, it would be valuable if the Authors could overlay their rearrangements data to Hi-C maps previously obtained from mouse thymocytes (for example, in Seitan et al., 2013), and explore whether frequently rearranged regions in thymic tumors overlap with specific TAD borders and/or are enriched in particular 3D compartments.

This is a very interesting suggestion, thank you very much. We have performed the requested analysis, and indeed, as can be now seen in Supp. Fig. 7, major regions of instability associated with *Atm*^{-/-} tumours lie within TAD boundaries. Although

only correlative, these results strengthen our conclusions (line 462-466).

-- In line 372, the Authors write that lesions can be stabilized upon ATM deficiency. It is not clear to me what is meant here: are the lesions (cleavage complexes) stabilized into full DSBs? In lines 84–85 the word stabilizing is used to describe the role of ATM in V(D)J recombination, where it stabilizes the post-cleavage complex and protects the DNA ends against incorrect ligation. Then, when speaking about ATM deficiency and in the context of lesion repair, this word choice is contradictory. The Authors should rewrite this sentence to make it less ambiguous.

Performed as requested.

-- Can the Authors comment on why the END-seq signal at TOP2B sites increases (rather than staying similar) in the Rag2^{-/-} thymocytes in Fig. 5D?

This is due to the fact that ENDseq signal at sites of direct RAG cleavage completely disappears. Since the technique is not completely quantitative, this redistribution of signal leads to an apparent increase at other sites. The Tcrb locus depicted in Fig. 7C, which harbours both RAG-dependent and -independent ENDseq peaks is an illustrative example for this. We have now clarified this in the text (lines 418-423).

-- The Authors should explain better in the method section why they use two different antibodies for TOP2B ChIPseq. Now the only place to find this information is in the Figure legend.

Performed as requested (lines 693-696).

-- In the methods section, the Authors should add that the ENCODE datasets they used were generated in mouse thymocytes. Furthermore, datasets from different sources and in some cases different cell types were used. This should be more explicitly mentioned in the manuscript, including the possible implications this may have.

Performed as requested (lines 708-714).

4) Analyses.

-- The Authors should be more explicit in their description of their analyses, particularly with regards to the NGS datasets. How are insulators and enhancers called and defined? How many peaks were called, and in the figures such as 4D, what exactly is a peak and what is not? Is the entire region bound by TOP2B or is everything between the grey regions considered background?

Performed as requested (316-320). In addition, we have been more careful

throughout the text when specifically referring to visible regions of accumulation (which may indeed harbour more than one peak) or to peaks called in the bioinformatics analysis.

-- Following the previous point, how are enriched regions defined in the END-seq data? For example in Fig. 5B, there are many peaks that, to the untrained eye, seem to be strong and enriched with respect to the WT, but they are not indicated with yellow. Furthermore, in line 364 and Fig. 5E the Authors regard the TOP2B signal at END-seq peaks, but how are peaks defined here, with what cut-off? To make the difference between the WT and the deficient thymocytes profiles more prominent, for example the significant differences should be marked by a different color, indicating differential peaks, or peak-called enriched regions should be more clearly indicated.

In order to avoid any ambiguity regarding ENDseq peaks, we have now only highlighted bioinformatically called peaks that accumulate in *Atm*^{-/-} thymocytes and that are not called in wild-type background. This is now indicated in the Figure Legends (lines 942-944), and appropriately referred to throughout the text.

-- The Authors should make a note about the variation of y axis ranges of the END-seq data. Comparison between the axes of Figure 7 and Figure 5 and Supp. Fig 5 shows huge differences, and the differences in Figure 5 seem very small compared to those in Supp. Fig 5. Maybe the axes in Fig. 5 can be extended, because the current view does not fit well with the description that 'no substantial END-seq signal was detected' (lines 351 and 352).

The accumulation of ENDseq signal at direct sites of RAG cleavage is much stronger than the widespread signal found elsewhere in the genome. In any case, as mentioned above, we have been more rigorous, displaying only *Atm*^{-/-} specific ENDseq peaks. Ranges have been adjusted in each region so the peak-calling differences can be appreciated.

-- The Authors should add an intersection of the END-seq data with the observed genome rearrangements in Figure 3 (or all the observed breakpoints in all animals as shown in the Supplementary Figure) to show that indeed, in the ATM^{-/-} compared to the WT, there is an enrichment of breaks in those regions (in a more quantitative manner than with genome browser shots of the most relevant genes).

As discussed above, and in the text, the resolution of the CGH analysis does not allow us to perform this type of analysis in a meaningful manner.

-- In Figure 5E, the TOP2B signal at END-seq peaks shows 'shoulders' in addition to the major center peak. The Authors should comment on these, perhaps by speculation.

The observed “shoulders” are also intriguing to us. Having in mind that TOP2 has to simultaneously bind two DNA segments in order to perform its function, they may represent the most abundant size of TOP2 mediated DNA loops. Alternatively, they could represent TOP2 recruitment to sites of DSBs to perform a yet-unknown function in signalling and/or repair. In any case, these hypotheses are too speculative and completely outside the scope of the present work, so we would not be comfortable with including them in the text.

-- Can the Authors quantify the genome-wide co-localization in percentage of peaks that overlap between TOP2B, RAG1, and RAG2, perhaps by extending the Venn diagram of Fig. 4, and refer to it in line 401?

We have now included the peak overlap between TOP2B and RAG as requested (Fig. 6B; lines 384-386).

-- How does Fig. 6C relate to Fig. 6A, the patterns of END-seq signal around RAG peaks in the WT and the two mutants seem different (more enriched in case of the ATM mutant) than can be observed in 6A?

Please note that the scales of the ENDseq heatmap in wild-type and *Atm*^{-/-} thymocytes in the original submission were different. We have now corrected this error (Fig. 6A).

Also, is there an explanation for why the END-seq signal goes up around RAG peaks when Rag2 is removed?

We understand that this can be puzzling. We believe that, as explained above for TOP2B sites, the apparent increase of ENDseq in Rag2^{-/-} may be due to the fact that much of the signal is relocated from the few but very strong sites of direct RAG cleavage to the very numerous regions of RAG binding in which DSBs are RAG independent (illustrated in Fig 7C).

-- To appreciate the few-kb displacement of DSB peaks in Figure 7, please add a zoom-in with a ruler to make this detail more appreciable.

Performed as requested (Supp. Fig. 7).

ADDITIONAL REMARKS

All corrected. Thank you very much for pointing them out.

-- The tile of the manuscript differs from the title on the Supplementary Information file.

-- In Fig. 2B the y-axis should be labeled Cumulative risk of developing thymic tumor to make the figure more directly comprehensible (same for all axes labeled Cumulative risk). Same holds true for other figures with this graph (e.g. risk of developing cancer in general, risk of developing thymic tumors).

-- In line 87, it is unclear what 'These defects in V(D)J recombination' refers to. In the alinea above this sentence, normal V(D)J recombination is described, including the role of ATM, and how RAG finds RSS sequences. The Authors should describe the defects they refer to in more detail or remove the word these.

-- Supplementary Figure 2A should be accompanied by a table indicating percentages of the respective precursors.

-- In line 187, a reference is made to Supp. Fig. 1D, but it needs to be adjusted to 1C.

-- Line 280 refers to Figure 3C, but there is no such figure in the manuscript I received.

-- All graphs should be given axes titles.

-- Figure 5 Legend does mention (D) twice, this should be correct to (D) and (E).

-- In line 420 the Authors refer to Supp. Fig. 4, stating that TOP2B accumulation in these regions is absent or largely decreased in MEFs. I don't agree with this statement, because in C the reduction is very minimal (nowhere near largely decreased).

-- In line 430 the Authors refer to Supp. Fig. 4B, which should most likely be corrected to 5B.

-- It is not clear to me why the Authors refer to the END-seq columns in Fig. 6A as RAG-dependent END-seq signal? Please check and/or clarify.

-- In the Discussion in line 549, reference is made to Figure 7 to show that TOP2B-RAG co-localization not only occurs at V(D)J regions but also genome-wide. Figure 7 shows no genome-wide co-localization as far as I see, but indeed the authors should add a figure showing that (a point that I have raised further above as well).

Reviewer #2 (Remarks to the Author): Expertise in TOP2B and genome architecture)

The authors report a strong combinatorial effect of loss of the DNA repair enzyme Tdp2 and ATM in the frequency of lymphoid cancers in a mouse model. These tumours are common in the ATM^{-/-} mouse model and in A-T patients and their aetiology is thought to involve inefficient and inaccurate repair of RAG-induced DSBs during antigen and T-cell receptor recombination. Since Tdp2 is required for a major mechanism for repair of TOP2-DNA adducts that may occur spontaneously in the genome, the authors conclude that endogenous TOP2 DNA lesions underlie the combined effect of Tdp2 and Atm loss. They also allude to the fact that part of the mechanism through which ATM leads to elevated tumour risk, especially in Tdp2 nulls could be via modulation of a nucleolytic pathway for TOP2-DNA adduct resolution (although this could be made much clearer). The first part of the paper describes the characteristics of Tdp2^{-/-}, ATM^{-/-} mice, the frequency of thymic tumour formation and genetic instability (CGH analysis) while the second part concerns TOP2B ChIP-seq analysis both globally and at specific loci. For this the authors performed their own TOP2B ChIP-seq in primary mouse thymocytes and interrogated this alongside available published data including End-seq data from Canela et al (2016) and Rag ChIP-seq from (Teng et al 2015). From these analyses the authors conclude that TOP2 occupancy colocalises to an extent with sites of spontaneous DNA breaks (End-seq) and with sites of RAG binding throughout the genome. In addition, the authors demonstrate that TOP2B is enriched at VDJ recombination regions in proximity to sites of RAD21 and RAG binding. At these sites there is a strong peak of endogenous DNA breaks (End-seq) which is RAG dependent. The authors conclude that their findings demonstrate a strong causal relationship between what they refer to as “accidental TO2B-induced DSBs” and cancer development.

Overall, I feel that the authors have produced some interesting and important findings relating to DNA damage and the sources of genome instability in the absence of clastogenic insult. The experiments appear to be well performed and the work is timely and significant. but there are number of major and minor issues that need addressing.

Major Issues

1. Page 8/Fig1 There is a lot made in this paper about increased accumulation of spontaneous TOP2 lesions and thus endogenous DNA damage in the absence of both ATM and Tdp2, but there is no direct evidence for this presented and this is a serious weakness. Elevated rates of “accidental” TOP2-mediated damage could perhaps be detected as increased numbers of “background” gH2AX foci in MEF cells, or ideally through comparing End-seq in WT, ATM^{-/-}, Tdp2^{-/-} cells and cells null for both.

We thank the reviewer for this comment, which we believe has made the

manuscript stronger. Indeed, as suggested, we now show that *Tdp2*^{-/-} *Atm*^{-/-} primary MEFs spontaneously accumulate 53BP1 foci during unchallenged cell growth (Fig. 7D). Furthermore, we have applied a novel technique called ICE-IP qPCR, by which we are now able to detect TOP2 covalently bound to specific genomic regions (Fig. 7E). With this, we have found a significant spontaneous enrichment of TOP2ccs and/or blocked TOP2 DSBs at two regions of the *Tcrb* locus in which RAG-independent DSBs accumulate (as determined by ENDseq). See lines 435-456.

2. Page 14, Fig 5. There is some significant over-interpretation here. Most seriously regarding the End-Seq data. It is stated that “the main peak of TOP2B-RAD21 binding at Notch was perfectly aligned with a region of increased End-seq signal” – the End-seq data in this region is “a choppy sea”, i.e. a continuous series of small peaks, possibly representing the noise in the system, and similar to the End-seq pattern for the other genes shown. The chances of one of these peaks lining up with a TOP2 peak by chance is clearly high, and so this data cannot be used as evidence of endogenous DNA damage concentrated at these TOP2B peaks. The same problem applies to the claim that in *ATM*^{-/-} cells peaks appear that coincide with the TOP2/RAD21 signal. Similarly, there are thousands of robust TOP2B peaks across the genome, about a third in promoters, so there is nothing special about the pattern of TOP2B distribution at the *Bcl11b*, *Pten* or *Notch1*, so it is not clear how this is connected with the genome instability evidenced by the CGH data.

As mentioned above in response to Reviewer 1, we are now more careful in considering ENDseq peaks in a more systematic fashion, and only highlight *Atm*^{-/-} specific peaks. In any case we have rephrased the entire section, specifically mentioned that part of the overlap may be due to the enrichment of TOP2B at regulatory regions (lines 345-348).

3. The source of the existing End-Seq and ChIP-seq data sets that were interrogated along with the TOP2B ChIP-seq data originating in this study must be acknowledged in the legends to Figs 4 and 5 at least, and possible also noted in the main Results text.

We have now been more careful to acknowledge the origin of datasets throughout the text, specifically clarifying that they were not generated in this study, and included specific references in the Figure Legends.

4. Line39, Fig 6: The possibility that the apparent overlap between TOP2/RAD21 peaks and RAG2 peaks could arise because they are both overrepresented in H3K4Me3 rich promoter regions should be considered.

This is now specifically mentioned in the text (lines 382-387).

5. The discussion is much too long and should be considerably shortened.

We have streamlined the Discussion, removing some considerations that were not essential, and, following Reviewer 1 suggestions, moving some of its content to Introduction.

6. More ChIP-seq/NGS details are required about the number of replicate samples, number of aligned reads etc and where the new data reported here (thymocyte Chip-seq) is deposited (although the latter may be something that is inserted after acceptance)

A link to the files containing all this information in the GEO database is now provided for review purposes, and will be release with the manuscript if accepted.

Minor Issues

All have been corrected. Thank you very much for pointing them out. Regarding point 21, we agree that the data, all though novel, does not add up from previous reports on TOP2B genome-wide localization. However, provided that we have not reached the limit of display items, we would prefer to keep it as a main Figure. Regarding point 22, this is just an example of a region in which two main TOP2B-binding sites can be simultaneously observed, corresponding to the major types of functional elements in which TOP2B is found: active regulatory regions and insulators.

1. Main manuscript and Supplemental material have different titles.
2. Line 81: Remove “being”
3. Line 84: Swap “thus” and “protecting”
4. Line 102-107: Confusing sentence, needs rewording
5. Line 108: for clarity I think “in ATM-/-“should be inserted between “in” and “mice”.
6. Line 118: RAG scanning is introduced here, but with no explanation of what it is, I think a brief description is required here for the benefit of the likely audience
7. Line 120: Is Ref 12 the correct one here, shouldn't it be Ref 13?
8. Line 122: Should the word “fully” be inserted between “not” and “suppress”
9. Line 130: replace “on” with “in”.
10. Page 6: Should also elude to nucleolytic processing pathway/s for resolving and repairing TOP2-DNA adducts involving MRN, CtIP and/or Artemis etc. This is an odd omission given that ATM could modulate this.
11. Page 6 bottom: Should give a reference for dietary factors such as flavonoids that could contribute to TOP2-DNA lesions.
12. Page 7 top: Given that there are frequent references to “accidental TOP2-induced DNA damage” consider mentioning here that existing endogenous DNA damage such as the presence of nicks or abasic sites may poison TOP2 in situ.
13. Page 155: Refs 41, but not Ref 42 shows this (42 just cites 41)
14. Page 7, 2nd para, see point 10 above, this is a bit lacking on mechanistically

how lack of ATM could lead to an accumulation of TOP2-DNA adducts and could also refer to work showing that ATLD cells exhibit elevated basal TOP2 complexes, consistent with a role for MRE11 in processing TOP2-DNA complexes.

15. Line 180: should be “neither did it aggravate”.

16. Fig 1B, y axis mislabelled, presumably it should be % of original weight rather than Weight loss (%).

17. Line 203-204: Confusingly worded

18. Line 216, Replace “in” with “over”

19. Line 282: should be ref 26??

20. Line 323: CTCF – typo

21. Fig 4. This could be moved into the Supplementary material, the data in this Fig is really just validating the thymocyte TOP2B ChIP-seq and doesn't show anything new or surprising.

22. Fig 4D, why was Cxcr4 in particular chosen?

23. Fig 5: There are two Ds and no E in the Figure legend

24. Fig 6C: End-seq signal at TOP2B/RAD21 peaks appears to be higher in Rag2^{-/-} cells than in WT, this is not discussed, but should be.

25. Line 434-435; The term “accidental TOP2 activity” used here and elsewhere is problematic. Presumably TOP2 has some real function at these sites, probably resolving supercoiling issues, so its activity is not accidental. I assume what is meant is incidental or unscheduled DNA damage resulting from (presumably) rare events where TOP2 fails to complete its reaction cycle leaving a stalled TOP2-DNA complex that is resolved either by the proteasome-Tdp2 or nucleolytic pathways. These instances need to be corrected and written more precisely.

Reviewer #3 (Remarks to the Author): Expertise in ATM and in vivo

This study by Alvarez-Quillon et al seeks to explore the causal relationship between TOP2-induced double strand breaks and cancer development focusing the attention in genome rearrangements in the absence of ATM kinase.

The elucidation of Topoisomerase functions and mechanisms in cellular systems deficient for DNA damage repair is relevant and of great interest for understanding the onset and progression of tumorigenesis.

The authors obtained a double knockout mouse model lacking of Atm kinase, the key regulator of DSBs repair and of TDP2 enzyme that unblocks TOP2-induced DSBs allowing the DNA repair. They show that double knockout mice have a phenotype with similarity with the Atm knockout mice in term of developmental defects and thymic genomic rearrangements.

The authors claim that compared to Atm knockout mice, the double knockouts display increased thymoma predisposition and are more sensitive to etoposide exposure. They have previously shown that etoposide stabilizes TOP2 cleavage complexes in double Atm and TDP2 knockout MEFs.

In support to their work the authors find enrichment of TOP2B in spontaneous DSBs of thymocytes with highest accumulation at the RAG sites and V(D)J-active regions.

Overall, the data in the present manuscript are insufficient to shed significant light on the topic and to support the conclusions of the authors, preventing the acceptance for publication.

The two major criticisms of the paper regard the comparative analysis of tumorigenesis between the double knockout mice and the *Atm* knockout model and the study limited to TOP2B isoform.

Comments below are provided as suggestions to improve this study.

Major points:

1. Other authors have shown that etoposide affects the survival of MEF cells obtained from *Atm* knockout embryos also at low concentration (Yamamoto K. et al., eLife 2016). A dose - response curve and quantification of etoposide damage should be done in double KO and *Atm* KO mice.

Yes, we have also shown before that *Atm*^{-/-} MEFs are mildly sensitive to etoposide, although substantially less than *Tdp2*^{-/-} and *Tdp2*^{-/-} *Atm*^{-/-} (Álvarez-Quilón 2014). Whether some sensitivity could perhaps be observed in *Atm*^{-/-} mice at higher doses of etoposide is outside the scope of this manuscript. We consider that the fact of *Tdp2*^{-/-} *Atm*^{-/-} mice being significantly sensitive to etoposide when compared to either of the single mutants is sufficiently proven.

2. The double knockout mice need further characterization to reach conclusions on their phenotype during development and cancer predisposition. Indeed B cells maturation and class switching recombination and meiotic differentiation must be evaluated in double KO compared to *Atm* KO mice.

We have now characterized B-cell precursors in the bone marrow of *Tdp2*^{-/-} *Atm*^{-/-} mice and have observed no significant differences with the single *Atm*^{-/-} (Supp. Fig 2D, lines 240-243). Although interesting, a characterization of class-switch recombination is outside the scope of this manuscript and could be considered for further studies. To address possible defects in meiosis we have also analysed histological sections of testes, finding no evident differences between *Tdp2*^{-/-} *Atm*^{-/-} and *Atm*^{-/-} animals (Supp. Fig. 1C, lines 170-171).

3. Figure 1 A and 1B need clarification to make conclusions on thymic lymphoma predisposition of double knockout mice vs *Atm* KO mice. The relative Kaplan-Meyer survival curves do not appear significant. In the figure legend should be reported how statistics has been performed. Statistic calculated only in the mean value of survival is incorrect. How was detected thymoma? Which is the

percentage of double KO and Atm KO mice that develop thymoma at different ages? (ie: at 3, 6, and 12 months). How the authors evaluated the cumulative risk? A description must be included in the methods section. Re-evaluation of these analyses is required also to state that latency of malignancy do not change in the two mouse models.

As a matter of fact, the differences in the Kaplan-Meier curves for overall survival and cumulative occurrence of thymic tumours are statistically significant as determined by the Wilcoxon test. We are aware that comparing mean values of survival is incorrect, but thank you for clarifying. Thymic tumours were identified macroscopically and validated by histopathological analysis. We have now changed the term “cumulative risk” to “cumulative occurrence”. We understand that this way it is more evident that the information regarding the percentage of animals that developed a tumour up to a given age is already contained within the graph. All this is now clearly indicated in the corresponding Figure Legend and in the Methods section.

4. The extensive genomic analysis done for TOP2B in wild-type thymocytes must be implemented with the analysis of TOP2A isomerase that is expressed in proliferating cells of the thymus that also undergo genomic rearrangements and it is associated with cancer predisposition.

Performed as requested (Supp. Fig. 4, lines 415-417).

Minor points:

All have been corrected. Thank you very much for pointing them out. Regarding point 2, we understand that this is due to the background, which has been shown to greatly affect the incidence of thymic lymphoma linked to ATM deficiency. For this reason all our experiments were performed with littermates. Testing ATM inhibitors *in vivo*, as suggested in point 3 is outside the scope of this study.

1. A comment on the different response observed in the small intestine of Atm KO and TDP2 KO mice exposed to etoposide should be inserted in the results.
2. Atm KO mice in standard housing conditions die, mainly for thymoma, between 4 and 6 months of age depending from the animal facilities, some of them reaching one year of age. The long life span (more than 1 year) reached by several AtmKO and double KO mice in the house condition described in the methods deserves attention and consideration to drive conclusions of in the study.
3. Authors should consider the use of Atm inhibitors *in vivo* to understand the structural Atm function in the repair of TOP2-induced DSBs.
4. Check the correspondence of the figures in the text. i.e. Fig 3b, Suppl fig 1C.
5. Check P value in figure legend 1B.

Reviewers' comments:

Reviewer #1 (Remarks to the Author):

The Authors have responded very well to all of the issues, which I raised in my first review. They have also added a relevant new technique (ICE-IP-qPCR) in support of their findings.

The flow and clarity of the revised manuscript have also improved compared to the first submission. There are still, however, several minor issues that, in my opinion, need to be addressed before the manuscript is accepted for publication. Please find them listed below.

REMARKS TO THE AUTHORS

-- In line 128 (and elsewhere in the manuscript), the term 'spontaneous TOP2 lesions' should be clarified; do the Authors refer to the incidence and impact of TOP2 lesions, when they are not induced by the TOP2 poison etoposide? If so, it would be good to clearly state that.

-- In line with the above point, it would be useful if the Authors gave a clear definition of 'spontaneous and/or accidental TOP2 lesions and DSBs'. In the revised manuscript these terms are used seemingly interchangeably, without a proper explanation of what is meant. Are 'accidental DSBs' those DSBs that are induced by TOP2, but not properly repaired? And what about 'spontaneous TOP2 lesions'? To prevent ambiguity, the Authors should either come up with a clear definition of the two types of lesions, or they should somehow circumvent the use of these terms. Related to this, can the Authors clarify the difference between 'spontaneous DSB accumulation' and 'regular DSB accumulation'?

-- Supplementary Figures: the legend for Suppl. Fig. 7 seems to correspond to Suppl. Fig. 6, and the numbering is off (there are 8 figures but 7 legends). Furthermore, is this Hi-C matrix truly genome-wide as suggested in the legend for Suppl. Fig. 7? Or are (pieces) of the three chromosomes shown? In the latter case, it might be clearer to simply show three separate matrices. Furthermore, the Authors should include the chromosome number and the positions along the chromosome for clarity.

-- For lines 140 - 145 it is unclear whether this is concluded from the Authors' previous paper, or whether it is a speculation. It would be good if this could be adjusted, in such a way that it becomes clear whether these are conclusions from the previous paper.

-- In Figure 5A, the highlight at the promoter region appears off (at least it does not overlap with the strong peak at the promoter)

-- Maybe rather than working with highlights to point attention to a certain peak that is absent/present, the Authors could add arrows to point specifically at the peak they describe in the text.

-- To clarify how ICE-IP-qPCR works, a schematic overview should be added to the Supplementary Figure panel.

-- In addition, more examples should be added of ICE-IP-qPCR on other regions that show different behavior, for example peaks from Figure 5.

-- The Methods section misses a description of the ENDseq data and how these were handled for analysis in this manuscript.

Reviewer #2 (Remarks to the Author):

Most issues that I raised in the initial version have been addressed, but there are still a number of points that need attention.

1. The first major issue that I raised with the initial assay concerned the lack of direct evidence for increased accumulation of spontaneous TOP2 lesions / DSBs in cells lacking Atm and Tdp2. This has now been partially addressed in Fig. 7D which shows an elevated proportion of cells with 53BP1 foci in Atm^{-/-}, TDP2^{-/-} cells. However, this data is near the end of the results section while at the beginning of the results (on line 175-178) it appears as a given, with no supporting evidence at this point, that the combined genotype will lead to increased DSB levels. Thus, it would seem a lot more logical to put this result (currently Fig 7D and possible 7E) at the beginning, either as (part of) a Figure or a supplementary Figure.

2. Figure 4. Parts are mislabelled conflicting between the text, legend and figure.

3. A bit more information or clearer description is needed to describe the differential peak calling on lines 340-345.

4. It is not clear to me how many replicates were performed for the ChIP-seq data reported, i.e. for TOP2B is the peak calling based on one or multiple replicates with each of the two antibodies employed? And for genome browser views are the traces based on single or merged multiple replicates? This information should be included in the Materials and Methods.

5. Lines 406-407 concerning the coincidence of TOP2B-cohesin peaks with End-seq peaks appears to be at odds with the data at least for Igh and Tcrb, where there is no coinciding End-seq signal in WT cells – the peak appears in Atm^{-/-} cells.

6. Discrepancy in the numbering of the supplementary figures – No legend for Sup Fig 5 and I think Sup figs 6 and 7 are the wrong way around.

7. Lines 413-415, it is important to have information about numbers of replica experiments before drawing comparative quantitative conclusions, and or to use statistically robust peak calling to determine the presence or absence of TOP2B peaks.

8. Line 462-466 the discussion of the hi-C data needs more explanation to help the reader interpret the result - regarding for example, how TAD borders are identified in these plots.

Minor point/s

9. Line 98, missing word after V(D)J??

10. Still not clear why the Cxcr4 gene in particular is used for the genome view illustration in Fig. 4

11. Fig. 5 A-C need to make it clearer in which orientation the genes are arranged

Reviewer #3 (Remarks to the Author):

The revised version of the manuscript by Alvarez-Quillon et al. has been implemented and improved. The text is more fluent and data are more adequately presented. In particular, the authors were able to present new results relative to the characterization of their mouse model and to demonstrate DNA damage accumulation in isolated cells from Tdp2 Atm double knockout mice. The major criticisms that I raised have been now addressed. Therefore, I recommend to accept this dataset for publication in Nature Communication.

We would like to thank the referees for their constructive comments. We have implemented all the suggested changes. Please find a point-by-point response below.

Reviewers' comments:

Reviewer #1 (Remarks to the Author):

The Authors have responded very well to all of the issues, which I raised in my first review. They have also added a relevant new technique (ICE-IP-qPCR) in support of their findings.

The flow and clarity of the revised manuscript have also improved compared to the first submission. There are still, however, several minor issues that, in my opinion, need to be addressed before the manuscript is accepted for publication. Please find them listed below.

REMARKS TO THE AUTHORS

-- In line 128 (and elsewhere in the manuscript), the term 'spontaneous TOP2 lesions' should be clarified; do the Authors refer to the incidence and impact of TOP2 lesions, when they are not induced by the TOP2 poison etoposide? If so, it would be good to clearly state that.

-- In line with the above point, it would be useful if the Authors gave a clear definition of 'spontaneous and/or accidental TOP2 lesions and DSBs'. In the revised manuscript these terms are used seemingly interchangeably, without a proper explanation of what is meant. Are 'accidental DSBs' those DSBs that are induced by TOP2, but not properly repaired? And what about 'spontaneous TOP2 lesions'? To prevent ambiguity, the Authors should either come up with a clear definition of the two types of lesions, or they should somehow circumvent the use of these terms. Related to this, can the Authors clarify the difference between 'spontaneous DSB accumulation' and 'regular DSB accumulation'?

Regarding these two points, we agree that the terminology we used may be confusing, thanks for pointing this out. We have now made an effort to clarify this issue. We have extended the Introduction on TOP2 induced DSBs (lines 108-134), and changed "spontaneous lesions" and "accidental lesions" to "endogenous lesions". We understand that this term is less ambiguous, and better represents the notion of not being exogenously induced by etoposide, which is our intention.

-- Supplementary Figures: the legend for Suppl. Fig. 7 seems to correspond to Suppl. Fig. 6, and the numbering is off (there are 8 figures but 7 legends). Furthermore, is this Hi-C matrix truly genome-wide as suggested in the legend for Suppl. Fig. 7? Or are (pieces) of the three chromosomes shown? In the latter case, it might be clearer to simply show three separate matrices. Furthermore, the Authors should include the chromosome number and the positions along the chromosome for clarity.

We have now corrected the errors in Figure Legends, thanks for pointing this out. Regarding Suppl. Fig. 7, the HiC matrixes correspond to chromosome regions. This is now clearly indicated in the figure and in the corresponding legend.

-- For lines 140 - 145 it is unclear whether this is concluded from the Authors' previous paper, or whether it is a speculation. It would be good if this could be adjusted, in such a way that it becomes clear whether these are conclusions from the previous paper.

We have clarified this issue (lines 140-140).

-- In Figure 5A, the highlight at the promoter region appears off (at least it does not overlap with the strong peak at the promoter)

The highlighted regions do not correspond to TOP2B peaks, but to ENDseq peaks that are specific to Atm-/- thymocytes. We hope we have now clarified this sufficiently in the main text (lines 346-349) and in the corresponding figure legend.

-- Maybe rather than working with highlights to point attention to a certain peak that is absent/present, the Authors could add arrows to point specifically at the peak they describe in the text.

We have now used arrows to specifically highlight the regions in which we have analyzed endogenous TOP2cc accumulation by ENDseq (see below) (Fig. 5 and Fig. 7).

-- To clarify how ICE-IP-qPCR works, a schematic overview should be added to the Supplementary Figure panel.

We agree that this can be helpful and have included it as Supp. Fig.6.

-- In addition, more examples should be added of ICE-IP-qPCR on other regions that show different behavior, for example peaks from Figure 5.

Performed as requested. We have selected two ENDseq positive regions in *Pten*, one with TOP2B signal and one without, and the results are now included as Fig. 7E and in the main text (lines 457-463).

-- The Methods section misses a description of the ENDseq data and how these were handled for analysis in this manuscript.

Performed as requested (lines 741-743).

Reviewer #2 (Remarks to the Author):

Most issues that I raised in the initial version have been addressed, but there are still a number of points that need attention.

1. The first major issue that I raised with the initial assay concerned the lack of direct evidence for increased accumulation of spontaneous TOP2 lesions / DSBs in cells lacking *Atm* and *Tdp2*. This has now been partially addressed in Fig. 7D which shows an elevated proportion of cells with 53BP1 foci in *Atm*^{-/-}, *TDP2*^{-/-} cells. However, this data is near the end of the results section while at the beginning of the results (on line 175-178) it appears as a given, with no supporting evidence at this point, that the combined genotype will lead to increased DSB levels. Thus, it would seem a lot more logical to put this result (currently Fig 7D and possible 7E) at the beginning, either as (part of) a Figure or a supplementary Figure.

We have now moved previous Fig. 7D to Fig. 1A, as well as the corresponding section in the main text (now lines 178-182). We agree that this arrangement has improved the flow of the manuscript, thank you.

2. Figure 4. Parts are mislabeled conflicting between the text, legend and figure.

Changed as requested, thank you.

3. A bit more information or clearer description is needed to describe the differential peak calling on lines 340-345.

Performed as requested. Information can be found in Fig. 5 legend and lines 346-349.

4. It is not clear to me how many replicates were performed for the ChIP-seq data reported, i.e. for TOP2B is the peak calling based on one or multiple replicates with each of the two antibodies employed? And for genome browser views are the traces based on single or merged multiple replicates? This information should be included in the Materials and Methods.

Performed as requested. Now included in Materials and Methods (lines 723-728).

5. Lines 406-407 concerning the coincidence of TOP2B-cohesin peaks with End-seq peaks appears to be at odds with the data at least for *Igh* and *Tcrb*, where there is no coinciding End-seq signal in WT cells – the peak appears in *Atm*^{-/-} cells.

We understand that this reflects the variable incidence of endogenous lesions and/or END seq detection at different sites. In some cases it might be enough to be detected in wildtype background, while in others stabilization due to ATM loss may be required.

6. Discrepancy in the numbering of the supplementary figures – No legend for Sup Fig 5 and I think Sup figs 6 and 7 are the wrong way around.

Changed as requested, thank you.

7. Lines 413-415, it is important to have information about numbers of replica experiments before drawing

comparative quantitative conclusions, and or to use statistically robust peak calling to determine the presence or absence of TOP2B peaks.

Information is now described in detail in Material and Methods and Supp. Fig. 4, and referred to in the main text (lines 420-424). Common and specific peaks are highlighted in different colors, as indicated.

8. Line 462-466 the discussion of the hi-C data needs more explanation to help the reader interpret the result - regarding for example, how TAD borders are identified in these plots.

We tried to clarify this issue (lines 530-534). We agree that we had been somewhat inaccurate when talking about TAD borders. We have now changed the text accordingly and talk about "regions of strong long-range interactions (in the order of megabases), apparent as borders of squares in the contact density map".

Minor point/s

All performed as requested

9. Line 98, missing word after V(D)J??

10. Still not clear why the Cxcr4 gene in particular is used for the genome view illustration in Fig. 4

11. Fig. 5 A-C need to make it clearer in which orientation the genes are arranged.

Reviewer #3 (Remarks to the Author):

The revised version of the manuscript by Alvarez-Quillon et al. has been implemented and improved. The text is more fluent and data are more adequately presented. In particular, the authors were able to present new results relative to the characterization of their mouse model and to demonstrate DNA damage accumulation in isolated cells from Tdp2 Atm double knockout mice. The major criticisms that I raised have been now addressed. Therefore, I recommend to accept this dataset for publication in Nature Communication.

REVIEWERS' COMMENTS:

Reviewer #1 (Remarks to the Author):

All my comments have been adequately addressed and the manuscript is now ready for publication.

Reviewer #2 (Remarks to the Author):

The authors have carried out various amendments and corrections to the previous version of the manuscript to take into account the issues that I raised previously. Each of those issues have been adequately addressed and in my opinion this work could now be accepted for publication. I just have two very minor points that require correction:

1: Supp Fig 8 is cited in the text before Supp Figs 6 and 7, and I think the order need to be corrected so that they are cited consecutively in order

2: The acronym "TAD" should be defined

We would like to thank the referees for their help and support. We have implemented the minor corrections suggested. Please find a point-by-point response below.

REVIEWERS' COMMENTS:

Reviewer #1 (Remarks to the Author):

All my comments have been adequately addressed and the manuscript is now ready for publication.

Reviewer #2 (Remarks to the Author):

The authors have carried out various amendments and corrections to the previous version of the manuscript to take into account the issues that I raised previously. Each of those issues have been adequately addressed and in my opinion this work could now be accepted for publication. I just have two very minor points that require correction:

1: Supp Fig 8 is cited in the text before Supp Figs 6 and 7, and I think the order need to be corrected so that they are cited consecutively in order

Performed as requested.

2: The acronym "TAD" should be defined

Performed as requested.